# Beyond Conflict: Subspace-Alignment as the Missing Piece of Model Merging

## Abstract

Model merging integrates task knowledge by combining task vectors (differences between pre-trained and fine-tuned weights) across models for both vision and large language models (LLMs). Existing methods improve merging performance solely by mitigating conflicts between task vectors. However, we find that conflict alone is not the full story: when conflict-oriented remedies are applied, severe interference still persists. This observation motivates a novel perspective: analysing the interference from the standpoint of alignment. Experiments reveal that task vectors exhibit high alignment in subspaces with large singular values. After merging, these aligned subspaces gather more singular values and produce activations with higher magnitude compared to others. The resulting spectral imbalance substantially degrades model performance. Inspired by this, in contrast to previous methods that focus solely on conflicts, we propose the Subspace-Alignment Aware Merging (AlignMerge), which quantifies alignment by projecting task vectors onto the shared singular subspaces of the merged task vector and attenuates overly aligned components. AlignMerge is training-free and requires no auxiliary data. Across vision and language benchmarks, it achieves state-of-the-art performance and narrows the gap to traditional multi-task learning to 3.6%.

## 1 Introduction

Model merging fuses trained models directly in weight space into a single model (Ruan et al., 2025). Rather than retraining models from scratch or relying on traditional model ensembling, model merging enables the direct manipulation and fusion of model weights to combine capabilities (Shah et al., 2025; Ortiz-Jimenez et al., 2023), forget detrimental knowledge (Ilharco et al., 2022; Ni et al., 2024), or mitigate catastrophic forgetting between tasks (Chitale et al., 2023; Marczak et al., 2025b). Model merging not only dramatically accelerates the development of large-scale models, such as large language models (LLMs) (Goddard et al., 2024; Wan et al., 2024), but also unlocks unprecedented flexibility in leveraging collective model intelligence (Huang et al., 2024; Oh et al., 2024). Current research primarily focuses on merging models that were fine-tuned on different tasks but share the same pre-trained backbone, yielding a single model with stronger multi-task capability. Furthermore, many of these works are based on the analysis and utilisation of task vectors (Ilharco et al., 2022), which are defined as the difference between pre-trained and fine-tuned weights.

Currently, most task-vector based methods attribute interference in model merging to a variety of conflict (*e.g.*, weight or singular conflict (Yadav et al., 2023; Gargiulo et al., 2025)). *Yet task vectors are nearly orthogonal (Ortiz-Jimenez et al., 2023), so large conflicts should be rare, yet interference persists, even when conflict-oriented remedies are applied.* Therefore, we propose a question: Is there any other factors beyond conflict, contribute to the interference? To uncover the underlying factors, we approach the problem from the opposite direction and analyse the **Alignment**. For clarity, we henceforth reshape each task vector into its matrix form. As illustrated in Fig. 1(b), an SVD of these task vectors reveals pronounced alignment in the top singular subspaces across both vision and language backbones (details in the Appendix). We term this phenomenon subspace-alignment. When task vectors are summed, components within the aligned subspaces add coherently, resulting in a substantial increase in the singular values of these subspaces relative to others (Fig. 1(c)). This, in turn, significantly amplifies activations of aligned subspaces during inference and degrades accuracy. The resulting spectral imbalance accumulation cautions against global scaling across subspaces and motivates a targeted remedy: attenuate overly aligned components in different subspaces.

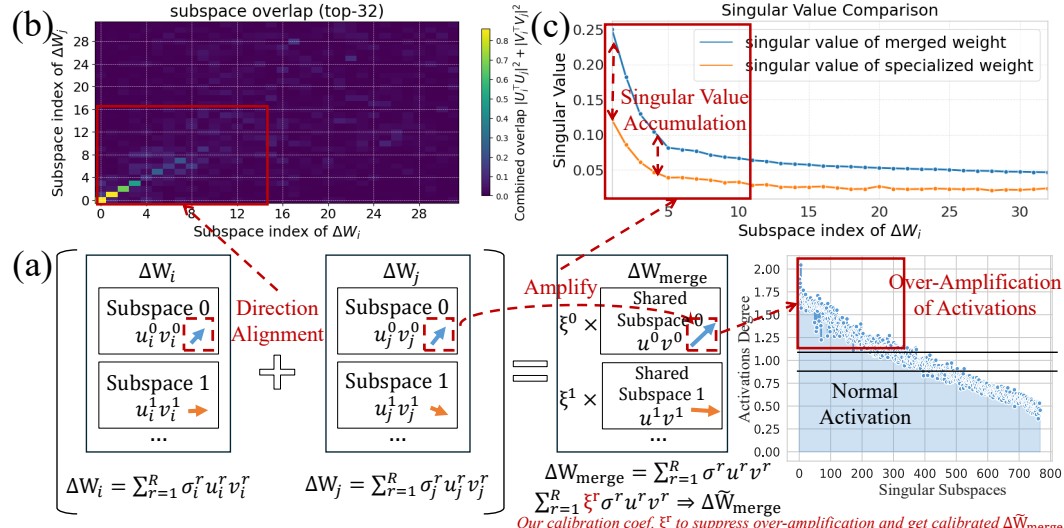

Figure 1: (a) Overview of the alignment across singular subspaces. Applying subspace calibration coefficients $\xi^r$ yields the corrected $\Delta\widetilde{\mathbf{W}}_{\text{merge}}$. (b) Similarity between two task vectors, measured by $S = \|U_A^\top U_B\|_F^2 + \|V_A^\top V_B\|_F^2$, where $U$ and $V$ are left/right singular vectors of the task vector. (c) Projecting the summed task vector onto each specialized task vector's subspace shows that stronger alignment yields greater singular-value accumulation within that subspace.

Specifically, in this paper, we focus on the shared singular subspaces obtained by applying SVD to the summed task vector $\Delta\mathbf{W}_{\text{merge}}$. We project any pair of task vector, $\Delta\mathbf{W}_i$ and $\Delta\mathbf{W}_j$, onto this common basis and compute a Subspace Cross-Influence $\xi_{i,j}^r$. Visualizing $\xi_{i,j}^r$ (Fig. 2) shows that subspaces with larger singular values tend to exhibit stronger alignment.

Motivated by these observations, we design two distinct *Subspace-alignment Aware Merging (Align-Merge)* strategies at pre/post-merging stages. At the **post-merging stage**, building on Marczak et al. (2025a) who observe gains from averaging singular values, *AlignMerge through Balancing (AlignMerge-B)* rescales each singular subspace using a calibration coefficient $\xi^r$ computed from pairwise Subspace Cross-Influence $\{\xi_{i,j}^r\}$. This subspace normalization suppresses over-amplification in highly aligned directions and improves overall accuracy.

At the **pre-merging stage**, by explicitly enforcing task vector orthogonality among all subspaces via whitening, *AlignMerge through orthogonalisation (AlignMerge-O)* can effectively eliminate the alignment of task vectos in different subspaces. Overall, AlignMerge-O substantially mitigates the activation over-amplification induced by subspace-alignment; however, by operating on weights in isolation, it limits the exploitation of shared structure and cross-task transfer. Extensive experiments validate the critical role of these effects and show that AlignMerge consistently achieves SOTA performance on diverse benchmarks. In summary, our research makes three key contributions:

1) We find that conflict explains only the tip of the interference iceberg; a substantial, overlooked portion arises from the alignment of task vectors within singular subspaces.

2) We propose two *Subspace-alignment Aware Merging (AlignMerge)* strategies at different stages, to effectively mitigate the over-amplification of activations caused by highly aligned subspaces.

3) Both theoretical analysis and experiments demonstrate the effectiveness of AlignMerge, reducing the performance gap to conventional MTL to just 3.6%, while also enabling targeted enhancement for specific tasks (*i.e.*, Preference Optimisation).

In this paper, Section 2 reviews related work. Section 3 formalizes subspace-alignment and analyses its impact on model merging. Building on these insights, Section 4 introduces two methods that mitigate alignment-induced interference.

## 2 RELATED WORK

**Dynamic Model Merging.** Unlike model ensembling, which combines the outputs or predictions of multiple independent models to improve generalisation (Dong et al., 2020), model merging operates directly at the weight level. In its common training-free (Zhang et al., 2025; Li et al., 2025; Yuan et al., 2025) form, it integrates the knowledge encoded in the parameters of several trained models into a single unified model (Yang et al., 2024a). This technology addresses challenges such as catastrophic forgetting (Chitale et al., 2023; Zhu et al., 2024; Marczak et al., 2025b), domain shift (Izmailov et al., 2019; Wortsman et al., 2022), and the efficient construction of LLMs (Dekoninck et al., 2024; Aiello et al., 2023). To reduce conflicts among models, a line of work studies dynamic merging, where the behaviour of the merged model depends on the input. For example, DaWin (Oh et al., 2024) performs input-conditioned interpolation, EMR-Merging (Huang et al., 2024) and TALL-Mask (Wang et al., 2024) learn task-specific masks, and Twin-Merging (Lu et al., 2024) introduces task experts in the spirit of mixture-of-experts. These approaches often achieve strong performance, but they rely on task or routing information at inference, which limits practicality.

**Static Model Merging.** Early research primarily focused on weight averaging or traditional interpolation strategies (Wortsman et al., 2022; Ilharco et al., 2022; Matena & Raffel, 2022; Jin et al., 2023), enabling rapid assembly of models with diverse task expertise but often suffering from suboptimal performance due to unresolved conflicts or redundancies among weights. Subsequent research has sought to mitigate inter-model conflicts under either data-dependent or data-free regimes. Data-dependent methods rely on auxiliary data such as validation sets or unlabeled test inputs. For example, PCB-Merging (Du et al., 2024) uses a validation set, NPS-Pruning (Du et al., 2025) employs a calibration set, and AdaMerging (Yang et al., 2024c) and Trust Region Arithmetic (Sun et al., 2025) perform test-time adaptation. Such dependencies reduce practical applicability in settings without accessible auxiliary data. Data-free methods can be grouped into weight-space and singular-vector perspectives. Weight-space methods (Yu et al., 2024; Yadav et al., 2023; He et al., 2024) localise and prune conflicting parameters to reduce incompatibility, while SVD-based approaches (Gargiulo et al., 2025; Stoica et al., 2024; Marczak et al., 2025a) focus on spectral alignment to reconcile conflicts at the subspace level. However, existing research focuses exclusively on various forms of conflict, overlooking the fact that subspace-alignment can also introduce inter-model interference and ultimately degrade performance.

## 3 WHAT CAUSE INTER-MODEL INTERFERENCE

Building on the preliminaries (Sec. 3.1), we formalize what subspace-alignment is and its measurement (Sec. 3.2), and then analyse how Subspace-Alignment cause interference (Sec. 3.3).

### 3.1 PRELIMINARY

Given a set of $K$ model parameter set $\{\mathbf{W}_i\}_{i=1}^K$, which fine-tuned on pre-trained parameter $\mathbf{W}_{\text{pre}}$. Model merging aims to construct a merged model $\mathbf{W}_{\text{merge}}$ that integrally inherits task-specific knowledge from all $\{\mathbf{W}_i\}$. Traditionally, a simple weight averaging is adopted: $\mathbf{W}_{\text{merge}} = \frac{1}{K}\sum_{i=1}^K \mathbf{W}_i$. Building on this, Task Arithmetic (TA) (Ilharco et al., 2022) introduces the concept of task vector $\Delta\mathbf{W}_i = \mathbf{W}_i - \mathbf{W}_{\text{pre}}$, and then applies a function $\mathcal{F}$ to merge task vectors, yielding $\Delta\mathbf{W}_{\text{merge}} = \mathcal{F}(\Delta\mathbf{W}_1, \Delta\mathbf{W}_2, \dots, \Delta\mathbf{W}_K)$. Finally, with an additional hyperparameter $\lambda$, model merging can be achieved in the following paradigm:

$$\mathbf{W}_{\text{merge}} = \mathbf{W}_{\text{pre}} + \lambda\Delta\mathbf{W}_{\text{merge}}. \tag{1}$$

However, this strategy fails to capture the inherent relationship between weights. Recent developments represent weight matrices in spectral form, typically via singular value decomposition (SVD) (Gargiulo et al., 2025):

$$\Delta\mathbf{W}_i = \mathbf{U}_i\mathbf{\Sigma}_i\mathbf{V}_i^\top = \sum_{r=1}^R \sigma_i^r \boldsymbol{u}_i^r (\boldsymbol{v}_i^r)^\top = \sum_{r=1}^R \sigma_i^r \mathbf{M}_i^r, \tag{2}$$

where $\mathbf{U}_i$ and $\mathbf{V}_i$ denote the left and right singular vectors, and $\mathbf{\Sigma}_i$ is a diagonal matrix of singular values. This decomposition can also be viewed as a weighted sum of $R$ outer products of singular vectors, which are the spans of the rank-one matrix $\mathbf{M}_i^r$ we called **singular vector subspace**.

For subsequent analysis, for 2 **rank-one matrices** $\mathbf{A}$ and $\mathbf{B}$, we define the Frobenius inner product-based projection coefficient $\mathbf{A}$ onto another matrix $\mathbf{B}$ as $\mathrm{Proj}_{\mathbf{B}}(\mathbf{A})$, which measures the alignment between two matrices and equals the projection between singular vectors (Details in Appendix B.1):

$$\mathrm{Proj}_{\mathbf{B}}(\mathbf{A}) = \frac{\langle \mathbf{A}, \mathbf{B} \rangle_F}{\langle \mathbf{B}, \mathbf{B} \rangle_F} = \frac{\mathrm{tr}(\mathbf{A}^\top \mathbf{B})}{||\mathbf{B}||_F^2}. \tag{3}$$

## 3.2 SUBSPACE-ALIGNMENT CAUSES INTERFERENCE

Model merging aims to combine the capabilities of specialized models into a single model. In practice, inter-model interference can cause these capabilities to cancel, so simple weight averaging often underperforms. Prior works largely attribute such interference to conflicts among task vectors (Yadav et al., 2023; Yu et al., 2024), yet severe degradation still remains after conflict-oriented remedies. This observation raises the question: Is there any other factors beyond conflict, contribute to the interference? We therefore examine the interference from the opposite direction, alignment.

As shown in Fig. 1(b), performing an SVD of each task vector in matrix form and comparing their left and right singular subspaces reveals a clear pattern: although task vectors are approximately orthogonal (Ilharco et al., 2022), their SVD-defined subspaces exhibit strong directional alignment. Further investigation of this alignment property demonstrates that stronger alignment in a subspace between the specialised models leads to larger singular-value growth of their merged model in that subspace, which inflates activations at inference. However, The alignment is highly non-uniform, high in the top singular subspaces and much weaker in lower ones (Fig. 1(b), Fig. 6). As a result, the accumulation of singular values across subspaces in the merged model varies significantly (Fig. 1(c)), leading to dominance of highly aligned subspaces and suppression of those with low alignment during inference. These observations motivate estimating alignment per subspace and applying targeted corrections to highly aligned components to prevent over-accumulation.

Based on this, model merging can be reformulated from the perspective of subspace. Specifically, let $\Delta \mathbf{W}_{\mathrm{merge}}$ denote the merged model, obtained by applying an arbitrary merging function to $K$ specialized models $\{\Delta \mathbf{W}_1, \ldots, \Delta \mathbf{W}_K\}$. Through singular value decomposition (SVD), $\Delta \mathbf{W}_{\mathrm{merge}}$ can then be represented as a sum over multiple shared subspaces $\mathbf{M}_{\mathrm{merge}}^r = \boldsymbol{u}_{\mathrm{merge}}^r (\boldsymbol{v}_{\mathrm{merge}}^r)^\top$, then we consider each subspace $\mathbf{M}_{\mathrm{merge}}^r$ as the projection of each specialized task vector $\{\Delta \mathbf{W}_k\}_{k=1}^K$ through projection matrices $\mathbf{P}_{\mathrm{col}}^r = \boldsymbol{u}_{\mathrm{merge}}^r (\boldsymbol{u}_{\mathrm{merge}}^r)^\top$ and $\mathbf{P}_{\mathrm{row}}^r = \boldsymbol{v}_{\mathrm{merge}}^r (\boldsymbol{v}_{\mathrm{merge}}^r)^\top$.

$$\Delta \mathbf{W}_{\mathrm{merge}} = \mathbf{U}_{\mathrm{merge}} \mathbf{\Sigma}_{\mathrm{merge}} \mathbf{V}_{\mathrm{merge}} = \sum_{r=1}^R \sigma_{\mathrm{merge}}^r \mathbf{M}_{\mathrm{merge}}^r = \sum_{r=1}^R \eta^r \sum_{i=1}^K \mathbf{P}_{\mathrm{col}}^r \Delta \mathbf{W}_i \mathbf{P}_{\mathrm{row}}^r, \tag{4}$$

where $\eta^r = \sigma_{\mathrm{merge}}^r / \left( (\boldsymbol{u}_{\mathrm{merge}}^r)^\top \sum_{i=1}^K \Delta \mathbf{W}_i \boldsymbol{v}_{\mathrm{merge}}^r \right)$ serves as a scaling factor. To assess the relationship between $\Delta \mathbf{W}_j$ and $\Delta \mathbf{W}_i$ within the shared subspace $\mathbf{M}_{\mathrm{merge}}^r$, and how this alignment amplify activation magnitudes at inference, we establish the following proposition, whereby the Subspace Cross-Influence $\xi_{i,j}^r$ of how task $j$ influence on task $i$ in shared subspace $r$ is derived.

---

**Proposition 1** (Subspace Cross-Influence) *Let $\Delta \mathbf{W}_i$ and $\Delta \mathbf{W}_j$ be 2 task vectors finetuned from different tasks, the extent that task $j$ influence on task $i$ within shared subspace $r$ can be calculated as Subspace Cross-Influence $\xi_{i,j}^r$ (The definition of $\mathbf{W}_{\mathrm{res}}^r$ and other details defer in Appendix B.2):*

$$\xi_{i,j}^r = \mathrm{Proj}_{\mathbf{P}_{\mathrm{col}}^r \Delta \mathbf{W}_i} \left( \mathbf{P}_{\mathrm{col}}^r \Delta \mathbf{W}_j \right) + \mathrm{Proj}_{\mathbf{W}_i^r \mathbf{P}_{\mathrm{row}}^r} \left( \mathbf{W}_{\mathrm{res}}^r \mathbf{P}_{\mathrm{row}}^r \right). \tag{5}$$

$$\mathbf{P}_{col}^r \Delta \mathbf{W}_j \mathbf{P}_{row}^r \approx \xi_{i,j}^r \mathbf{P}_{col}^r \Delta \mathbf{W}_i \mathbf{P}_{row}^r \tag{6}$$

---

For Subspace Cross-Influence $\xi_{i,j}^r$, a positive value indicates alignment between tasks $i$ and $j$ in subspace $r$. The overlapping component of $\Delta \mathbf{W}_j$ then adds coherently to $\Delta \mathbf{W}_i$ at inference (*i.e.*,

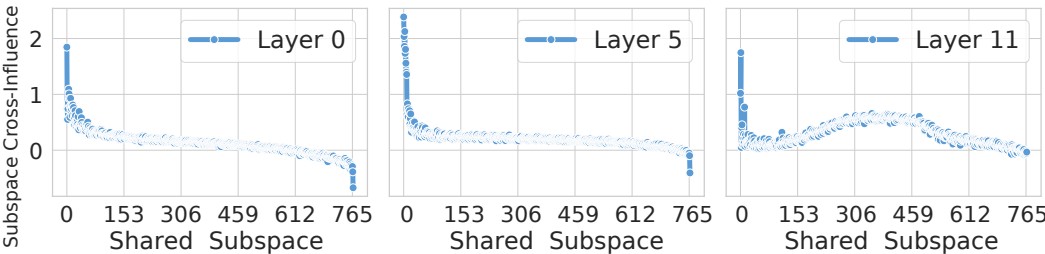

Figure 2: Visualisation of Subspace Cross-Influence of layer 'mlp.c_proj.weight', illustrating the impact of MNIST on Cars. Notably, subspace with lower index correspond to larger singular values.

"double counting"), amplifying activations in that subspace. We visualize the $\xi_{i,j}^r$ across different subspaces in Fig. 2. The pattern mirrors Fig. 6: positive alignment is concentrated in shared subspaces with large singular values and decays as the singular values decrease.

### 3.3 HOW SUBSPACE-ALIGNMENT CAUSE INTERFERENCE

After computing the Subspace Cross-Influence coefficients $\xi_{i,j}^r$, the whole influence on target task $i$ in subspace $r$ is $\sum_{j\neq i} \xi_{i,j}^r$. By combining with Eq.(4), the merged task vector in the shared subspace $r$ can be represented as the projection of the task vector only corresponding to task $i$:

$$\sigma_{\text{merge}}^r \mathbf{M}_{\text{merge}}^r \approx \eta^r \left( \sum_{j=1,\,j\neq i}^{K} \xi_{i,j}^r + 1 \right) \mathbf{P}_{\text{col}}^r \, \Delta\mathbf{W}_i \, \mathbf{P}_{\text{row}}^r \tag{7}$$

**Proposition 2** (Optimal subspace coefficient) *Let* $\mathbf{S} = \sum_{r=1}^{R} \xi^r \mathbf{P}_{\text{col}}^r \Delta\mathbf{W}_i^r \mathbf{P}_{\text{row}}^r$, *where* $\xi^r$ *denotes the full coefficient in Eq. 7 before* $\mathbf{P}_{col}^r$. *Then* $\|\Delta\mathbf{W}_i - \mathbf{S}\|_F$ *is minimised if* $\xi^r = 1$ *for all* $r$.

**Proof 2** *The above problem can be formulated as the following optimisation problem:*

$$\min_{\mathbf{\Sigma}} \; \|\Delta\mathbf{W}_i - \mathbf{S}\|_F^2, \quad \text{s.t.} \quad \mathbf{S} = \mathbf{U}_{\text{merge}} \, \mathbf{\Sigma} \, \mathbf{V}_{\text{merge}}, \quad \mathbf{\Sigma} \text{ is diagonal.}$$

*This problem admits a closed-form solution (Details in Appendix B.3), given by*

$$\mathbf{\Sigma} = \text{diag}\left( \mathbf{U}_{\text{merge}}^{\top} \, \Delta\mathbf{W}_i \, \mathbf{V}_{\text{merge}} \right), \qquad \mathbf{\Sigma}^r = \boldsymbol{u}_{\text{merge}}^{\top} \, \Delta\mathbf{W}_i \, \boldsymbol{v}_{\text{merge}}.$$

*When* $\xi^r = 1$, *the singular value corresponding to rank* $r$ *is exactly* $\boldsymbol{u}_{\text{merge}}^{\top} \mathbf{W}_i \, \boldsymbol{v}_{\text{merge}}$, *thereby ensuring optimal similarity. Therefore, the optimal choice is* $\xi^r = 1$ *for all* $r$.

By Proposition 2, as the full coefficient $\xi^r$ approaches one, the distance $\|\Delta\mathbf{W}_{\text{merge}} - \Delta\mathbf{W}_i\|_F$ decreases. We therefore define the *Subspace-Alignment Index (SAI)* as the absolute deviation of the full coefficient from one:

$$SAI_i^r = \left| \eta^r \left( \sum_{j=1,\,j\neq i}^{K} \xi_{i,j}^r + 1 \right) - 1 \right|. \tag{8}$$

Lower $SAI_i^r$ values indicate less interference for task $i$ within subspace $r$. We visualize the signed $SAI_i^r$ (without absolute value) for representative baselines in Fig. 3 on ViT-B/32 and Llama2-7B. Simple weight averaging (WA) reduces interference in top singular subspaces but over-suppresses bottom subspaces with weaker alignment. Task Arithmetic (TA) assigns larger scaling coefficients to task vectors, preventing over-suppression in the bottom subspaces and better balancing corrections across the spectrum, thereby reducing overall interference. TIES mitigates conflicts via weight-space pruning, likewise driving $SAI_i^r$ closer to zero.

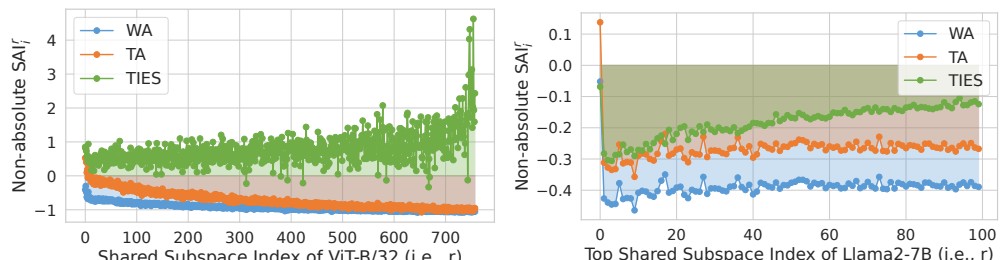

Figure 3: Visualization of non-absolute $\text{SAI}_i^r$ across subspaces, indicating the influence of all other models on the target; values closer to $0$ denote lower interference. Left: ViT-B/32. Right: Llama2.

# 4 METHODOLOGY

As shown above, alignment within singular subspaces can degrade the merged model's performance. We propose two complementary strategies. In post-merging stage, AlignMerge-B rescales each shared subspace using the Subspace Cross Influence (Sec. 4.1). In pre-merging stage, AlignMerge-O orthogonalizes the task vector subspaces to remove shared alignment prior to combination (Sec. 4.2).

## 4.1 SUBSPACE-ALIGNMENT AWARE MERGING THROUGH BALANCING (ALIGNMERGE-B)

At the **post-merging stage**, the most direct way to eliminate the impact of subspace-alignment on a specific task $i$ is to divide the merged model's singular value $\sigma_{merge}^r$ in Eq. (7) by the full coefficient $\tilde{\xi}_i^r = \eta^r(\sum_{j=1, j \neq i}^{K} \xi_{i,j}^r + 1)$, thereby ensuring $SAI_i^r$ equals to zero.

To control the strength of the calibration, we introduce a gating hyperparameter $\alpha$. Specifically, we rescale only those subspaces with $\tilde{\xi}_i^r > \alpha$. For $\tilde{\xi}_i^r \leq 0$, which indicates that $\Delta \mathbf{W}_i$ is non-contributory in that subspace, we set the corresponding component to zero. Formally,

$$\Delta \widetilde{\mathbf{W}}_{\text{merge}} = \sum_{r=1}^{R} \frac{\sigma_{\text{merge}}^r}{d(\tilde{\xi}_i^r)} \mathbf{M}_{\text{merge}}^r, \quad d(\tilde{\xi}_i^r) = \begin{cases} \tilde{\xi}_i^r, & \text{if } \tilde{\xi}_i^r \geq \alpha \\ 1, & \text{if } 0 \leq \tilde{\xi}_i^r < \alpha \\ \infty. & \text{if } \tilde{\xi}_i^r \leq 0 \end{cases} \quad (9)$$

When aiming to improve the performance of multiple tasks simultaneously, $\tilde{\xi}^r$ can be set as the average of $\tilde{\xi}_i^r$ across different tasks $i$, and substituted for $\tilde{\xi}_i^r$ in Eq. (9) accordingly. The final weights are then obtained by $\mathbf{W}_{\text{merge}} = \mathbf{W}_{\text{pre}} + \Delta \widetilde{\mathbf{W}}_{\text{merge}}$.

## 4.2 SUBSPACE-ALIGNMENT AWARE MERGING THROUGH ORTHOGONALISATION (ALIGNMERGE-O)

At the **pre-merging stage**, subspace-alignment can also be eliminated by orthogonalizing singular subspaces across different specialised models. Consequently, Eq. (4) can be rewritten as

$$\sigma_{\text{merge}}^r \mathbf{M}_{\text{merge}}^r = \eta^r \mathbf{P}_{\text{col}}^r \Delta \mathbf{W}_{i^*}^r \mathbf{P}_{\text{row}}^r \text{ where } i^* = \arg\max_i \|\mathbf{P}_{\text{col}}^r \Delta \mathbf{W}_i^r \mathbf{P}_{\text{row}}^r\|_F \quad (10)$$

In this case, each singular subspace is contributed by a single model, which removes cross-model alignment and yields $\text{SAI}_i^r = 0$. This also explains the effectiveness of TSV-M (Gargiulo et al., 2025). However, enforcing fully orthogonality limits the exploitation of shared structure and weakens cross-task transfer. Instead, we retain and inject a selected set of aligned subspaces into the merged model, while correcting their singular magnitudes with the scaling factor $\eta^r$.

As shown in Fig. 2, the top singular vectors of weights from different tasks generally exhibit strong alignment. To avoid information loss in these strongly aligned subspaces caused by forced orthogonalisation, we first explicitly extract their aligned subspace. Specifically, we perform SVD on

each task-specific weight matrix, obtaining $\mathbf{U}_i = [\ldots, \boldsymbol{u}_i^r, \ldots, \boldsymbol{u}_i^R], \boldsymbol{\Sigma}_i = [\ldots, \sigma_i^r, \ldots, \sigma_i^R], \mathbf{V}_i = [\ldots, \boldsymbol{v}_i^r, \ldots, \boldsymbol{v}_i^R]$. We retain the top rank-1 singular triplets to form aligned singular vectors $\boldsymbol{u}_{\text{top}}$, $\boldsymbol{v}_{\text{top}}$ and aligned singular value $\sigma_{top}$:

$$\boldsymbol{u}_{\text{top}} = \frac{\sum_{i=1}^{K} \boldsymbol{u}_i^1}{||\sum_{i=1}^{K} \boldsymbol{u}_i^1||_2}, \quad \sigma_{top} = \frac{1}{K} \sum_{i=1}^{K} \sigma_i^1, \quad \boldsymbol{v}_{\text{top}} = \frac{\sum_{i=1}^{K} \boldsymbol{v}_i^1}{||\sum_{i=1}^{K} \boldsymbol{v}_i^1||_2}. \tag{11}$$

After extracting the aligned subspace, we need to retain the residual subspace of each specialised model with respect to the aligned subspace to preserve task-specific information. Specifically, we compute the rank-one matrices $\mathbf{M}_{\text{top}} = \sigma_{top} \boldsymbol{u}_{\text{top}} (\boldsymbol{v}_{\text{top}})^\top$ and $\mathbf{M}_i^1 = \sigma_i^1 \boldsymbol{u}_i^1 (\boldsymbol{v}_i^1)^\top$. And then project $\mathbf{M}_i^1$ onto $\mathbf{M}_{\text{top}}$ to obtain the residual component $\mathbf{M}_{\text{res},i}$.

$$\mathbf{M}_{\text{res},i} = \mathbf{M}_i^1 - \text{Proj}_{\mathbf{M}_{\text{top}}}(\mathbf{M}_i^1)\mathbf{M}_{\text{top}}. \tag{12}$$

Applying SVD on $\mathbf{M}_{\text{res},i}$ yields its top singular vectors and value: $\boldsymbol{u}_{\text{res},i}$, $\sigma_{\text{res},i}$ and $\boldsymbol{v}_{\text{res},i}$. We then concatenate $\boldsymbol{u}_{\text{top}}$, $\{\boldsymbol{u}_{\text{res},i}\}_{i=1}^K$ and $\{\boldsymbol{u}_i^r\}_{i=1}^K$ for subspaces from 2 to $L = (R-1)/K$ to construct $\hat{\mathbf{U}}$. The same procedure is applied to $\hat{\boldsymbol{\Sigma}}$ and $\hat{\mathbf{V}}$. The formal process is given as follows:

$$\hat{\mathbf{U}} = [\boldsymbol{u}_{\text{top}}, \{\boldsymbol{u}_{\text{res},i}, \boldsymbol{u}_i^{2:L}\}_{i=1}^K], \hat{\mathbf{V}} = [\boldsymbol{v}_{\text{top}}, \{\boldsymbol{v}_{\text{res},i}, \boldsymbol{v}_i^{2:L}\}_{i=1}^K], \hat{\boldsymbol{\Sigma}} = [\boldsymbol{\sigma}_{\text{top}}, \{\boldsymbol{\sigma}_{\text{res},i}, \boldsymbol{\sigma}_i^{2:L}\}_{i=1}^K], \tag{13}$$

here $\boldsymbol{u}_i^{2:L}$ denotes the collection $\boldsymbol{u}_i^r$ for $r = 2, \ldots, L$, and similarly for the other terms. Following TSV-M (Gargiulo et al., 2025), we apply whitening to orthogonalise both $\hat{\mathbf{U}}$ and $\hat{\mathbf{V}}$ and obtain orthogonal bases $\hat{\mathbf{U}}_\perp$ and $\hat{\mathbf{V}}_\perp$. Specifically, we perform SVD on $\hat{\mathbf{U}}$ as $\hat{\mathbf{U}} = \mathbf{P}\mathbf{D}\mathbf{Q}^\top$, and set $\hat{\mathbf{U}}_\perp = \mathbf{P}\mathbf{Q}^T$. The computation for $\hat{\mathbf{V}}_\perp$ is analogously.

Finally, we reconstruct the merged weight $\Delta\widetilde{\mathbf{W}}_{\text{merge}}$ using the singular vectors $\hat{\mathbf{u}}_\perp^r$ and $\hat{\mathbf{v}}_\perp^r$ from $\hat{\mathbf{U}}$ and $\hat{\mathbf{V}}$, respectively, and divide by $\eta$ to ensure that the distance between the merged model and the individual models in the weight space is minimised. Pseudocode is provided in Algorithm 1.

$$\Delta\widetilde{\mathbf{W}}_{\text{merge}} = \sqrt{\frac{\sum_{r=1}^{R}(\hat{\sigma}^r)^2}{\sum_{r=1}^{R}(\hat{\sigma}^r/\eta^r)^2}} \sum_{r=1}^{R} \frac{\hat{\sigma}^r}{\eta^r} \hat{\mathbf{u}}_\perp^r (\hat{\mathbf{v}}_\perp^r)^\top. \tag{14}$$

When computing the final merged model, we additionally multiply by the factor $\sqrt{\frac{\sum_{r=1}^{R}(\hat{\sigma}^r)^2}{\sum_{r=1}^{R}(\hat{\sigma}^r/\eta^r)^2}}$ to ensure that the overall Frobenius norm of the weights remains consistent before and after scaling by $\eta^r$ (Details in Appendix B.4). The final weights are then obtained by $\mathbf{W}_{\text{merge}} = \mathbf{W}_{\text{pre}} + \Delta\widetilde{\mathbf{W}}_{\text{merge}}$.

## 5 EXPERIMENTS

### 5.1 EXPERIMENTAL PROTOCOL

**Baselines and Datasets.** For empirical validation, we benchmark our method against several SOTA training-free model merging methods, including TA (Ilharco et al., 2022), TIES Yadav et al. (2023), DARE (Yu et al., 2024), TSV-M (Gargiulo et al., 2025) and Iso-C (Marczak et al., 2025a). All results are obtained from our own experiments under a unified evaluation protocol. As the checkpoints used here may differ from those in the original works, the reported performance may deviate from that in its original paper. Notably, current training-free methods have outperformed training-based methods like AdaMerging++ (Yang et al., 2024c) and Surgery (Yang et al., 2024b).

Consistent with prior studies Ilharco et al. (2022); Yang et al. (2024c), our primary experiments regard computer vision (CV) are conducted on 8 classification benchmarks, and primary experiments regard natural language processing (NLP) are conducted on 11 classification benchmarks and 2 open LLM Leaderboard. More detail can be found in Appendix.

**Implementation Details.** We use the ViT-B/32 CLIP model as the default visual encoder, consistent with the setup in Yang et al. (2024c). By default, the scaling coefficient $\lambda = 0.3$ for both Task Arithmetic (TA) and TIES-Merging (TIES), as recommended in their papers. For parameter pruning, TIES and DARE preserve the top $20\%$ of the parameters. The hyperparameters introduced in AlignMerge-B can be selected based on the validation set, we use default values of $\alpha = 1.8$.

## 5.2 VISION & LANGUAGE: MAIN RESULTS

**Computer Vision (CV) Experiments**. Following prior work(Marczak et al., 2025a), we evaluate average classification accuracy across 8 and 14 datasets and show in Tab. 1 (Details are defer to Appendix E.1). Both AlignMerge-B and AlignMerge-O consistently achieve SOTA results in diverse merging tasks. In particular, the performance gap between AlignMerge-O and the empirical upper bound of traditional MTL methods (88.9 % for ViT-B/16) is reduced to only 3.6%.

Table 1: Consolidated average accuracy (%) across CV benchmarks. Per-dataset results are deferred to the appendix. Due to the use of different checkpoints, certain methods (*e.g.*, Iso-C and Iso-CTS) exhibit differently from reported in the original paper.

| Method | 8 Tasks | | | 14 Tasks | | |
|---|---|---|---|---|---|---|
| | **ViT-B/32** | **ViT-B/16** | **ViT-L/14** | **ViT-B/32** | **ViT-B/16** | **ViT-L/14** |
| Reference (non-merging) | | | | | | |
| Pretrained | 48.0 | 55.2 | 64.9 | 56.6 | 61.7 | 70.4 |
| Individual | 90.5 | 93.0 | 94.4 | 87.3 | 89.5 | 91.4 |
| Training-free merging | | | | | | |
| TA (Ilharco et al., 2022) | 69.1 | 73.8 | 84.4 | 64.0 | 64.3 | 72.9 |
| TIES (Yadav et al., 2023) | 72.2 | 75.7 | 84.4 | 61.6 | 61.6 | 78.7 |
| DARE (Yu et al., 2024) | 65.8 | 71.5 | 79.4 | 63.9 | 67.2 | 76.5 |
| TSV-M (Gargiulo et al., 2025) | 84.0 | 87.3 | 91.5 | 76.4 | 77.0 | 86.1 |
| Iso-C (Marczak et al., 2025a) | 83.1 | 87.5 | 91.4 | 73.4 | 72.3 | 86.7 |
| Iso-CTS (Marczak et al., 2025a) | 81.4 | 86.9 | 90.9 | 76.7 | 79.2 | **87.5** |
| AlignMerge-B (Ours) | 84.9 | 88.5 | **92.2** | 76.8 | 77.4 | 86.5 |
| AlignMerge-O (Ours) | **85.3** | **88.7** | 92.1 | **78.5** | **79.3** | 86.8 |

**Natural Language Processing (NLP) Experiments.** We evaluate our method across NLP models of varying sizes in Table 2 (Details defer to Appendix E.1). AlignMerge achieves SOTA performance on both conventional small language models and recent large language models (LLMs). For T0, we follow Yu et al. (2024) and adopt IA[3]-based Parameter-efficient fine-tuning (PEFT). Because IA[3] yields task vectors as vectors rather than full weight matrices, an SVD is not defined; consequently, SVD-dependent approaches (*e.g.*, TSV-M, Iso-C, AlignMerge-O) are not applicable, denoted as —.

Table 2: Consolidated performance across NLP benchmarks. Details are deferred to the appendix. Llama2 is evaluated on two generation benchmarks; others are used as encoders for classification.

| Method | Generative Evaluation | | Encoder-derived Classification | | |
|---|---|---|---|---|---|
| | **Llama2-7B (FT)** | | **BERT (FT)** | **T5 (FT)** | **T0 (PEFT)** |
| | **AlphcaEval↑** | **GSM8K↑** | **Avg Acc (%)** | **Avg Acc (%)** | **Avg Acc (%)** |
| TA (Ilharco et al., 2022) | 42.6 | 43.8 | 67.0 | 41.5 | 62.2 |
| TIES (Yadav et al., 2023) | 40.3 | 45.0 | 64.3 | 41.2 | 62.8 |
| DARE (Yu et al., 2024) | 37.2 | 47.0 | 65.8 | 45.5 | 65.1 |
| TSV-M (Gargiulo et al., 2025) | 48.1 | 49.7 | 69.3 | 46.5 | — |
| Iso-C (Marczak et al., 2025a) | 29.5 | 42.0 | 63.7 | 43.9 | — |
| Iso-CTS (Marczak et al., 2025a) | 24.0 | 38.7 | 62.5 | 38.8 | — |
| **AlignMerge-B** (Ours) | 48.3 | 47.9 | **70.6** | **50.0** | **65.9** |
| **AlignMerge-O** (Ours) | **88.1** | **51.0** | 69.8 | 46.6 | — |

### 5.3 EMPIRICAL ANALYSIS AND ABLATION STUDIES

**Ablation Study**. We ablate the sensitivity-control hyperparameter $\alpha$ in AlignMerge-B. As shown in Fig. 4, $\alpha$ thresholds the degree of subspace-alignment that triggers correction. Even with $\alpha = 0$ (no gating), AlignMerge-B consistently outperforms baselines such as TA. With proper tuning of $\alpha$, AlignMerge-B delivers further gains and surpasses the current state of the art. However, setting $\alpha$ too high degrades performance because few subspaces are corrected. For unseen tasks, our empirical results suggest using $\alpha \approx \frac{1}{4}|\mathcal{T}|$ as a reasonable default, where $|\mathcal{T}|$ denotes the number of tasks.

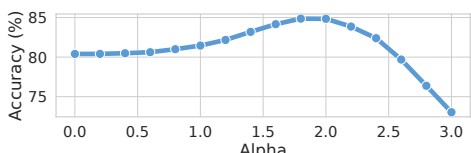

Figure 4: Ablation alpha on AlignMerge-B.

Table 3: Ablation study on AlignMerge-O.

| Method | Removed component | Acc. (%) |
|---|---|---|
| AlignMerge-O | None | **85.32** |
| | w/o Alignment Extraction | 85.11 |
| | w/o Divide $\eta$ | 84.21 |
| | w/o Both Components | 83.99 |

For AlignMerge-O, which has no hyperparameters, we assess the contribution of each component in the setting with eight tasks (Tab. 3). The results show that the extraction of alignment subspaces and the normalization by $\eta$ are both essential for effective model merging. Additional experiments are provided in Appendix E.2.

**Preference Optimisation**. AlignMerge-B can suppresses over-amplification of activations for specific tasks, thereby improving task-specific performance. We refer to this, which accentuates particular model performance post-merging, as Preference Optimisation. In Fig. 5, each cell at position $(i, j)$ indicates the performance in task $j$ after preference optimisation specifically for task $i$ (with column-wise normalisation). The numbers in parentheses indicate the original classification accuracy. It can be observed that preference optimisation of AlignMerge-B typically yields high values for the diagonal elements in each column. Additionally, related tasks may benefit from this optimisation, while those with large domain differences may see degraded performance. For

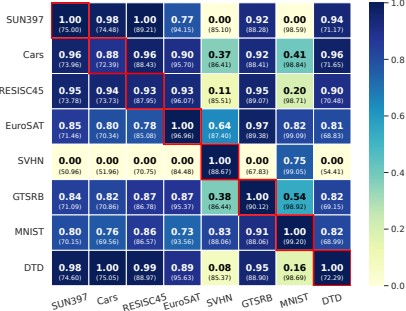

Figure 5: Merging performance target on 1 task.

example, optimising for "Cars" improves performance on the natural image dataset "SUN397", whereas remote sensing ("EuroSAT") and digit recognition ("SVHN") tasks show limited gains.

**Cost Analysis**. Consistent with TSV-M and ISO-CTS, our method applies SVD to a selected subset of linear layers, which introduces additional cost. Even on 7B-parameter LLMs, this overhead is acceptable. Moreover, the overall cost is substantially lower than training-based multi-task learning (MTL), since no parameter updates are performed and the decompositions are computed once offline. Table 4 summarizes the runtime and peak memory footprint of AlignMerge-O across backbones.

Table 4: Additional runtime and memory usage caused by our merging across backbones.

| Backbone | Time Cost | Memory Usage |
|---|---|---|
| ViT-B/32 | 131.4 s | 8.21 MiB |
| ViT-B/16 | 130.9 s | 8.21MiB |
| ViT-L/14 | 422.8 s | 8.22MiB |
| LLaMA2 7B | 1899.9s | 8.19MiB |

## 6 CONCLUSION

We revisit interference in model merging beyond conflict-only explanations. By analyzing weights across singular subspaces, we show that top singular subspaces exhibit strong alignment that induces activation over-amplification at inference and degrades performance. To address this, we introduce two independent strategies: AlignMerge-B, which applies subspace-aware rescaling after merging, and AlignMerge-O, which enforces subspace orthogonalization before merging while preserving aligned structure. Both approaches consistently mitigate over-alignment and achieve SOTA results across vision and language benchmarks, with especially strong gains on LLMs.

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

# A LLM USAGE

We used a large language model solely as a writing assistant for copy-editing—improving grammar, wording, and clarity. The LLM did not generate or modify any technical content (ideas, algorithms, equations, proofs, experimental design, code, results, or figures). All scientific claims and analyses were conceived and written by the authors.

# B DETAILED PROOF

## B.1 PROOF THAT THE PROJECTION FOR RANK-ONE MATRICES REDUCES TO SINGULAR VECTOR PROJECTION.

The projection coefficient of $\mathbf{A}$ onto $\mathbf{B}$ in Frobenius inner product space is given by

$$\text{Proj}_{\mathbf{B}}(\mathbf{A}) = \frac{\langle \mathbf{A}, \mathbf{B} \rangle_F}{\langle \mathbf{B}, \mathbf{B} \rangle_F} = \frac{\text{tr}(\mathbf{A}^\top \mathbf{B})}{\|\mathbf{B}\|_F^2}. \tag{15}$$

Let $\mathbf{A} = \mathbf{a}_1 \mathbf{a}_2^\top \in \mathbb{R}^{m \times n}$ and $\mathbf{B} = \mathbf{b}_1 \mathbf{b}_2^\top \in \mathbb{R}^{m \times n}$ be two rank-one matrices, where $\mathbf{a}_1, \mathbf{b}_1 \in \mathbb{R}^m$ and $\mathbf{a}_2, \mathbf{b}_2 \in \mathbb{R}^n$. By substituting the forms of $\mathbf{A}$ and $\mathbf{B}$, we have

$$\mathbf{A}^\top \mathbf{B} = (\mathbf{a}_2 \mathbf{a}_1^\top)(\mathbf{b}_1 \mathbf{b}_2^\top) = \mathbf{a}_2 (\mathbf{a}_1^\top \mathbf{b}_1) \mathbf{b}_2^\top, \tag{16}$$

$$\text{tr}(\mathbf{A}^\top \mathbf{B}) = (\mathbf{a}_1^\top \mathbf{b}_1) \, \text{tr}(\mathbf{a}_2 \mathbf{b}_2^\top) = (\mathbf{a}_1^\top \mathbf{b}_1)(\mathbf{b}_2^\top \mathbf{a}_2). \tag{17}$$

Similarly,

$$\|\mathbf{B}\|_F^2 = \text{tr}(\mathbf{B}^\top \mathbf{B}) = \text{tr}\left((\mathbf{b}_2 \mathbf{b}_1^\top)(\mathbf{b}_1 \mathbf{b}_2^\top)\right) \tag{18}$$

$$= \text{tr}\left(\mathbf{b}_2 (\mathbf{b}_1^\top \mathbf{b}_1) \mathbf{b}_2^\top\right) = (\mathbf{b}_1^\top \mathbf{b}_1)(\mathbf{b}_2^\top \mathbf{b}_2). \tag{19}$$

Therefore, the projection coefficient can be written as:

$$\text{Proj}_{\mathbf{B}}(\mathbf{A}) = \frac{(\mathbf{a}_1^\top \mathbf{b}_1)(\mathbf{b}_2^\top \mathbf{a}_2)}{(\mathbf{b}_1^\top \mathbf{b}_1)(\mathbf{b}_2^\top \mathbf{b}_2)} = \frac{\langle \mathbf{a}_1, \mathbf{b}_1 \rangle}{\|\mathbf{b}_1\|^2} \cdot \frac{\langle \mathbf{a}_2, \mathbf{b}_2 \rangle}{\|\mathbf{b}_2\|^2} \tag{20}$$

That is, the projection coefficient of a rank-one matrix onto another rank-one matrix under the Frobenius inner product reduces to the product of the projection coefficients of their singular vectors.

## B.2 PROOF OF PROPOSITION 1.

Considering the relationship between the merged model $\Delta\mathbf{W}_{\text{merge}}$ and 2 specialised models $\Delta\mathbf{W}_i, \Delta\mathbf{W}_j$, we project $\Delta\mathbf{W}_j$ to assess its actual impact on task $i$:

$$\sigma_{\text{merge}}^r \mathbf{M}_{\text{merge}}^r = \eta^r \mathbf{P}_{\text{col}}^r \Delta\mathbf{W}_i \mathbf{P}_{\text{row}}^r + \eta^r \mathbf{P}_{\text{col}}^r \Delta\mathbf{W}_j \mathbf{P}_{\text{row}}^r \tag{21}$$

We begin by projecting $\mathbf{P}_{\text{col}}^r \Delta\mathbf{W}_j^r$ onto the subspace spanned by $\mathbf{P}_{\text{col}}^r \Delta\mathbf{W}_i^r$:

$$\mathbf{P}_{\text{col}}^r \Delta\mathbf{W}_j^r \mathbf{P}_{\text{row}}^r \mathbf{x}_i = \alpha_1 \mathbf{P}_{\text{col}}^r \Delta\mathbf{W}_i^r \mathbf{P}_{\text{row}}^r \mathbf{x}_i + \mathbf{W}_{\text{res}}^r \mathbf{P}_{\text{row}}^r \mathbf{x}_i, \tag{22}$$

where $\alpha_1 = \text{Proj}_{\mathbf{P}_{\text{col}}^r \Delta\mathbf{W}_i^r} \left( \mathbf{P}_{\text{col}}^r \Delta\mathbf{W}_j^r \right)$, and $\mathbf{W}_{\text{res}}^r$ is the residual component orthogonal to $\mathbf{P}_{\text{col}}^r \Delta\mathbf{W}_i^r$. Next, we project $\mathbf{W}_{\text{res}}^r \mathbf{P}_{\text{row}}^r$ onto the subspace spanned by $\mathbf{W}_i^r \mathbf{P}_{\text{row}}^r$:

$$\mathbf{W}_{\text{res}}^r \mathbf{P}_{\text{row}}^r \mathbf{x}_i = \alpha_2 \mathbf{W}_i^r \mathbf{P}_{\text{row}}^r \mathbf{x}_i + \mathbf{W}_{\text{noise}}^r \mathbf{x}_i, \tag{23}$$

where $\alpha_2 = \text{Proj}_{\mathbf{W}_i^r \mathbf{P}_{\text{row}}^r} \left( \mathbf{W}_{\text{res}}^r \mathbf{P}_{\text{row}}^r \right)$, and $\mathbf{W}_{\text{noise}}^r$ is the final orthogonal residual.

Here, we estimate the total coefficient as

$$\xi_{i,j}^r = \alpha_1 + \alpha_2 \tag{24}$$

and the remaining residual $\epsilon_{i,j}^r = \mathbf{P}_{\text{col}}^r \mathbf{W}_{\text{noise}}^r$ as noise term. This leads to the expression in Proposition 1.

## B.3 PROOF OF CLOSED-FORM SOLUTION OF THE OPTIMISATION PROBLEM IN PROPOSITION 2.

We provide a proof for the closed-form solution to the optimisation problem stated in Proposition 2. The goal is to solve:

$$\min_{\boldsymbol{\Sigma}} \|\boldsymbol{\Delta}\mathbf{W}_i - \mathbf{S}\|_F^2, \quad \text{s.t.} \quad \mathbf{S} = \mathbf{U}_{\text{merge}} \boldsymbol{\Sigma} \mathbf{V}_{\text{merge}}, \quad \boldsymbol{\Sigma} \text{ is diagonal.} \tag{25}$$

The objective function can be expanded as follows:

$$J(\boldsymbol{\Sigma}) = \|\boldsymbol{\Delta}\mathbf{W}_i - \mathbf{U}_{\text{merge}} \boldsymbol{\Sigma} \mathbf{V}_{\text{merge}}\|_F^2 \tag{26}$$
$$= \text{trace}\left( (\boldsymbol{\Delta}\mathbf{W}_i - \mathbf{U}_{\text{merge}} \boldsymbol{\Sigma} \mathbf{V}_{\text{merge}})^\top (\boldsymbol{\Delta}\mathbf{W}_i - \mathbf{U}_{\text{merge}} \boldsymbol{\Sigma} \mathbf{V}_{\text{merge}}) \right) \tag{27}$$
$$= \text{trace}(\boldsymbol{\Delta}\mathbf{W}_i^\top \boldsymbol{\Delta}\mathbf{W}_i) - 2\,\text{trace}(\boldsymbol{\Sigma} \mathbf{V}_{\text{merge}} \boldsymbol{\Delta}\mathbf{W}_i^\top \mathbf{U}_{\text{merge}}) + \text{trace}(\boldsymbol{\Sigma}^\top \boldsymbol{\Sigma}). \tag{28}$$

By defining $M = \mathbf{U}_{\text{merge}}^\top \boldsymbol{\Delta}\mathbf{W}_i \mathbf{V}_{\text{merge}}$, the objective simplifies to:

$$J(\boldsymbol{\Sigma}) = \text{const} - 2\,\text{trace}(\boldsymbol{\Sigma}^\top M) + \text{trace}(\boldsymbol{\Sigma}^2). \tag{29}$$

Given that $\boldsymbol{\Sigma} = \text{diag}(\sigma_1, \sigma_2, \ldots, \sigma_r)$ and $M = [m_{ij}]$ with $m_{ii} = m_i$, the objective becomes:

$$J(\boldsymbol{\Sigma}) = \text{const} - 2\sum_i \sigma_i m_i + \sum_i \sigma_i^2. \tag{30}$$

Taking the derivative with respect to each $\sigma_i$ and setting it to zero gives:

$$\frac{\partial J}{\partial \sigma_i} = -2m_i + 2\sigma_i = 0 \quad \Rightarrow \quad \sigma_i = m_i. \tag{31}$$

Thus, the optimal solution is:

$$\boldsymbol{\Sigma}^* = \text{diag}(\mathbf{U}_{\text{merge}}^\top \boldsymbol{\Delta}\mathbf{W}_i \mathbf{V}_{\text{merge}}). \tag{32}$$

### B.4 Proof that the Factor in Eq. 11 Ensures Frobenius Norm Invariance

We prove that the normalisation factor in Eq. 11 guarantees that the Frobenius norm remains invariant before and after scaling.

Let the original Frobenius norm of the weight matrix $\mathbf{W}$ be:

$$\|\mathbf{W}\|_F^2 = \sum_{r=1}^{R} (\sigma^r)^2. \tag{33}$$

After scaling the singular values $\sigma^r$ by factors $\eta^r$, the new Frobenius norm becomes:

$$\left\| \sum_{r=1}^{R} \frac{\sigma^r}{\eta^r} \boldsymbol{u}_1^r (\boldsymbol{v}_1^r)^\top \right\|_F^2 = \sum_{r=1}^{R} \left( \frac{\sigma^r}{\eta^r} \right)^2. \tag{34}$$

To maintain the Frobenius norm, we introduce the normalisation factor: $\sqrt{\frac{\sum_{r=1}^{R} (\sigma^r)^2}{\sum_{r=1}^{R} \left( \frac{\sigma^r}{\eta^r} \right)^2}}$.

Thus, the Frobenius norm of the scaled weights is:

$$\left\| \widetilde{\mathbf{W}}_{\text{merge}} \right\|_F^2 = \left( \sqrt{\frac{\sum_{r=1}^{R} (\sigma^r)^2}{\sum_{r=1}^{R} \left( \frac{\sigma^r}{\eta^r} \right)^2}} \right)^2 \cdot \sum_{r=1}^{R} \left( \frac{\sigma^r}{\eta^r} \right)^2 = \sum_{r=1}^{R} (\sigma^r)^2. \tag{35}$$

Therefore, the normalisation factor ensures that the Frobenius norm of the weights remains unchanged after scaling, thereby preserving the representational capacity of the merged model.

## C Pseudocode of AlignMerge-O

We provide the pseudocode for AlignMerge-O, where the intermediate orthogonalisation steps are similar to those in TSV-M (Gargiulo et al., 2025). The detailed pseudocode is shown in Algorithm 1.

## D Additional Implementation Setting

**Datasets**. Consistent with prior studies Ilharco et al. (2022); Yang et al. (2024c), our primary experiments are conducted on 8 image classification benchmarks with various domain shift: SUN397 Xiao et al. (2016), Cars Krause et al. (2013), RESISC45 Cheng et al. (2017), EuroSAT Helber et al. (2019), SVHN Netzer et al., GTSRB Stallkamp et al. (2011), MNIST MNI, and DTD Cimpoi et al. (2014). To further demonstrate the versatility of our method, we extend our evaluation to some additional datasets: Caltech101 Fei-Fei et al. (2004), CIFAR10 Krizhevsky, CIFAR100 Krizhevsky, FGVC Maji et al. (2013), Flowers102 Nilsback & Zisserman (2008), Food101 Bossard et al. (2014), OxfordPets Parkhi et al. (2012), STL10 Coates et al. (2011), PCAM Veeling et al. (2018), FER2013 Goodfellow et al. (2013), EMNIST Cohen et al. (2017), FashionMNIST Xiao et al. (2017), RenderedSST2 Socher et al. (2013) and KMNIST Clanuwat et al. (2018). We fine-tune BERT on four binary classification datasets: AG News (Zhang et al., 2015), Rotten Tomatoes (Pang & Lee, 2005), CoLA (Warstadt et al., 2019), and SMS (Almeida et al., 2011). The resulting models are merged into a unified model using model merging techniques and evaluated on each task individually. TA and TIES use default settings, while TSV-M and AlignMerge-O are applied exclusively to BERT's most critical linear layer ("output.dense.weight"). Finally, to evaluate the performance of the merged model on LLMs, the fine-tuned models WizardMath-7B-V1.0 (Luo et al., 2023) and Llama-2-7b-chat-hf (Touvron et al., 2023) are merged and tested on two benchmarks, AlpacaEval (Li et al., 2023) and GSM8K (Cobbe et al., 2021).

**Datasets License**. Datasets distributed under the MIT License include: SVHN Netzer et al., STL10 Coates et al. (2011), EMNIST Cohen et al. (2017), FashionMNIST Xiao et al. (2017), and KMNIST Clanuwat et al. (2018).

Datasets released under various Creative Commons licenses consist of: EuroSAT Helber et al. (2019), DTD Cimpoi et al. (2014), RESISC45 Cheng et al. (2017), Food101 Bossard et al. (2014)

---

**Algorithm 1:** AlignMerge through orthogonalisation (AlignMerge-O)

---

**Input:** Pretrained weight $\Delta\mathbf{W}_{\text{pre}}$ and $K$ task-specific weights $\Delta\mathbf{W}_1, \ldots, \Delta\mathbf{W}_K$

**Output:** Merged model weight $\mathbf{W}_{\text{merge}}$

1 **for** $i = 1$ **to** $K$ **do**

2     Compute SVD: $\Delta\mathbf{W}_i = \mathbf{U}_i\boldsymbol{\Sigma}_i\mathbf{V}_i^\top = \sum_{r=1}^{R}\sigma_i^r \boldsymbol{u}_i^r(\boldsymbol{v}_i^r)^\top = \sum_{r=1}^{R}\sigma_i^r\mathbf{M}_i^r$ ;

3     Retain top $\frac{R-1}{K}$ singular components of $\mathbf{U}_i, \boldsymbol{\Sigma}_i,$ and $\mathbf{V}_i$;

4 **Extract the aligned subspace:** ;

5 $\boldsymbol{u}_{\text{top}} = \frac{\sum_{i=1}^{K}\boldsymbol{u}_i^1}{||\sum_{i=1}^{K}\boldsymbol{u}_i^1||_2}, s_{top} = \frac{1}{K}\sum_{i=1}^{K}s_i^1, \boldsymbol{v}_{\text{top}} = \frac{\sum_{i=1}^{K}\boldsymbol{v}_i^1}{||\sum_{i=1}^{K}\boldsymbol{v}_i^1||_2}$ ;

6 $\mathbf{M}_{\text{top}} = \sigma_{top}\boldsymbol{u}_{\text{top}}(\boldsymbol{v}_{\text{top}})^\top$;

7 **Compute the residual component:** ;

8 **for** $i = 1$ **to** $K$ **do**

9     $\mathbf{M}_i^1 = \sigma_i^1\boldsymbol{u}_i^1(\boldsymbol{v}_i^1)^\top$ ;

10     $\mathbf{M}_{\text{res},i} = \mathbf{M}_i^1 - \left(\text{Proj}_{\mathbf{M}_{\text{top}}^1}(\mathbf{M}_i^1)\right)\mathbf{M}_{\text{top}}$ ;

11     **Retain only main component in** $\mathbf{M}_{\text{res},i}$**:** ;

12     $\mathbf{M}_{\text{res},i} = \mathbf{U}_{\text{res},i}\boldsymbol{\Sigma}_{\text{res},i}(\mathbf{V}_{\text{res},i})^\top = \sum_{r=1}^{R}\sigma_{\text{res},i}^r \boldsymbol{u}_{\text{res},i}^r(\boldsymbol{v}_{\text{res},i}^r)^\top$ ,;

13 **Concatenate the matrices:** ;

14     $\hat{\mathbf{U}} = \left[\boldsymbol{u}_{\text{top}}, \{\boldsymbol{u}_{\text{res},i}^1\}_{i=1}^K, \{\boldsymbol{u}_i^{2:(R-1)/K}\}_{i=1}^K\right]$ ;

15     $\hat{\boldsymbol{\Sigma}} = \left[\boldsymbol{\sigma}_{\text{top}}, \{\boldsymbol{\sigma}_{\text{res},i}^1\}_{i=1}^K, \{\boldsymbol{\sigma}_i^{2:(R-1)/K}\}_{i=1}^K\right]$ ;

16     $\hat{\boldsymbol{V}} = \left[\boldsymbol{v}_{\text{top}}, \{\boldsymbol{v}_{\text{res},i}^1\}_{i=1}^K, \{\boldsymbol{v}_i^{2:(R-1)/K}\}_{i=1}^K\right]$ ;

17 **Compute** the SVD of $\hat{\mathbf{U}}$ and $\hat{\mathbf{V}}$: ;

18 $\hat{\mathbf{U}} = \mathbf{P}_U\mathbf{D}_U\mathbf{Q}_U^\top$;

19 $\hat{\mathbf{V}} = \mathbf{P}_V\mathbf{D}_V\mathbf{Q}_V^\top$;

20 **Obtain the orthogonal matrices:** ;

21 $\hat{\mathbf{U}}_\perp = \mathbf{P}_U\mathbf{Q}_U^\top = [\hat{\boldsymbol{u}}_\perp^1, \hat{\boldsymbol{u}}_\perp^2, \ldots, \hat{\boldsymbol{u}}_\perp^R]$;

22 $\hat{\mathbf{V}}_\perp = \mathbf{P}_V\mathbf{Q}_V^\top = [\hat{\boldsymbol{v}}_\perp^1, \hat{\boldsymbol{v}}_\perp^1, \ldots, \hat{\boldsymbol{v}}_\perp^R]$;

23 **for** $r = 1$ **to** $R$ **do**

24     $\eta^r = \hat{\sigma}^r / \left((\hat{\boldsymbol{u}}_\perp^r)^\top \sum_{i=1}^K \Delta\mathbf{W}_i\hat{\boldsymbol{v}}_\perp^r\right)$

25 **Reconstruct** the merged matrix: ;

26 $\Delta\widetilde{\mathbf{W}}_{\text{merge}} = \sqrt{\frac{\sum_{r=1}^{R}(\hat{\sigma}^r)^2}{\sum_{r=1}^{R}(\hat{\sigma}^r/\eta^r)^2}} \sum_{r=1}^{R}\frac{\hat{\sigma}^r}{\eta^r}\hat{\boldsymbol{u}}_\perp^r(\hat{\boldsymbol{v}}_\perp^r)^\top$;

27 **Construct** merged model weights: ;

28 $\mathbf{W}_{\text{merge}} = \mathbf{W}_{\text{pre}} + \Delta\widetilde{\mathbf{W}}_{\text{merge}}$;

29 **return** $\mathbf{W}_{\text{merge}}$ ;

---

The following datasets are available strictly for non-commercial research or academic use, typically under custom or restrictive academic licenses: SUN397 Xiao et al. (2016), Cars Krause et al. (2013), GTSRB Stallkamp et al. (2011), FGVC Maji et al. (2013), Flowers102 Nilsback & Zisserman (2008), OxfordPets Parkhi et al. (2012), Caltech101 Fei-Fei et al. (2004), FER2013 Goodfellow et al. (2013), PCAM Veeling et al. (2018), and RenderedSST2 Socher et al. (2013).

The MNIST MNI and CIFAR10/100 Krizhevsky datasets are provided for unrestricted research use and are considered to be in the public domain or distributed without explicit license restrictions.

For full details regarding dataset licenses and terms of use, please refer to the official web pages or documentation of the respective datasets.

**Implementation Detail**. All experiments are conducted using PyTorch on a single NVIDIA GeForce A800 GPU. Following prior work (Ilharco et al., 2022; Yadav et al., 2023), the value of $\alpha$ can be selected via the validation set: $\alpha = 0.6$ for two-model merging tasks, $\alpha = 2.8$ for fourteen-model tasks, and $\alpha = 6.5$ for twenty-model tasks.

# E  EXPERIMENTS

Here, we present experiments that were not included in the main paper due to space limitations.

## E.1  EXPERIMENTS DETAILS

**Merging experiments on 8 CV benchmark** The performance of each method on individual datasets is presented in detail. Complete results for ViT-B/32, ViT-B/16, and ViT-L/14 are provided in Table 5, Table 6, and Table 7, respectively.

| Method | SUN397. | Cars. | RESISC45. | EuroSAT. | SVHN. | GTSRB. | MNIST. | DTD. | Avg Acc. |
|---|---|---|---|---|---|---|---|---|---|
| Pretrained | 62.3 | 59.7 | 60.7 | 45.5 | 31.4 | 32.6 | 48.5 | 43.8 | 48.1 |
| Individual | 75.3 | 77.7 | 96.1 | 99.7 | 97.5 | 98.7 | 99.7 | 79.4 | 90.5 |
| Traditional MTL | 73.9 | 74.4 | 93.9 | 98.2 | 95.8 | 98.9 | 99.5 | 77.9 | 89.1 |
| TA | 64.7 | 63.3 | 71.4 | 72.7 | 64.2 | 52.8 | 87.5 | 50.1 | 65.8 |
| DARE | 64.8 | 63.5 | 71.8 | 72.4 | 63.8 | 52.4 | 87.5 | 50.5 | 65.8 |
| TIES | 59.6 | 58.6 | 71.0 | 81.3 | 86.1 | 70.9 | 98.4 | 54.7 | 72.6 |
| TSVM | 69.1 | 70.7 | 85.5 | 94.3 | 92.0 | 91.9 | 99.3 | 69.2 | 84.0 |
| ISO-C | 74.8 | 74.1 | 87.9 | 92.9 | 83.1 | 86.0 | 98.2 | 67.9 | 83.1 |
| ISO-CTS | 74.4 | 74.4 | 87.2 | 90.4 | 76.8 | 83.3 | 97.4 | 67.0 | 81.4 |
| AlignMerge-B | 73.7 | 73.8 | 88.8 | 95.6 | 86.7 | 89.7 | 98.9 | 71.5 | 84.8 |
| AlignMerge-O | 72.0 | 74.2 | 86.9 | 94.7 | 91.0 | 92.7 | 99.2 | 71.8 | **85.3** |

Table 5: Performance comparison across different methods and 8 datasets on ViT-B/32.

| Method | SUN397. | Cars. | RESISC45. | EuroSAT. | SVHN. | GTSRB. | MNIST. | DTD. | Avg Acc. |
|---|---|---|---|---|---|---|---|---|---|
| Pretrained | 63.8 | 64.7 | 66.4 | 54.6 | 52.0 | 43.4 | 51.7 | 44.7 | 55.2 |
| Individual | 81.8 | 86.8 | 96.9 | 99.8 | 97.9 | 99.2 | 99.8 | 82.1 | 93.0 |
| TA | 61.1 | 65.9 | 74.0 | 76.2 | 88.0 | 73.9 | 98.4 | 53.0 | 73.8 |
| DARE | 67.6 | 70.0 | 76.1 | 78.6 | 75.3 | 59.8 | 94.4 | 50.1 | 71.5 |
| TIES | 59.1 | 72.5 | 80.5 | 84.0 | 85.0 | 71.5 | 98.1 | 54.9 | 75.7 |
| TSVM | 72.8 | 80.4 | 89.1 | 96.6 | 93.9 | 93.9 | 99.3 | 72.7 | 87.3 |
| ISO-C | 78.1 | 82.3 | 91.9 | 96.9 | 88.3 | 91.9 | 98.8 | 71.9 | 87.5 |
| ISO-CTS | 77.9 | 83.2 | 92.1 | 96.4 | 84.9 | 91.3 | 98.4 | 71.1 | 86.9 |
| AlignMerge-B | 76.3 | 81.0 | 91.7 | 97.6 | 92.1 | 94.2 | 99.1 | 75.8 | 88.5 |
| AlignMerge-O | 75.4 | 82.9 | 90.7 | 98.0 | 93.0 | 95.0 | 99.2 | 75.6 | **88.7** |

Table 6: Performance comparison across different methods and 8 datasets on ViT-B/16.

| Method | SUN397. | Cars. | RESISC45. | EuroSAT. | SVHN. | GTSRB. | MNIST. | DTD. | Avg Acc. |
|---|---|---|---|---|---|---|---|---|---|
| Pretrained | 66.9 | 77.9 | 71.3 | 62.2 | 58.5 | 50.6 | 76.4 | 55.4 | 64.9 |
| Individual | 84.9 | 92.4 | 97.4 | 99.7 | 98.1 | 99.2 | 99.7 | 84.2 | 94.4 |
| TA | 74.0 | 81.9 | 86.5 | 92.9 | 87.8 | 86.8 | 99.0 | 66.1 | 84.4 |
| DARE | 71.1 | 81.6 | 82.6 | 90.6 | 78.3 | 70.8 | 97.0 | 63.1 | 79.4 |
| TIES | 74.4 | 84.5 | 88.0 | 94.0 | 85.7 | 82.1 | 98.7 | 67.7 | 84.4 |
| TSVM | 79.1 | 89.9 | 94.0 | 98.9 | 95.3 | 96.2 | 99.5 | 79.1 | 91.5 |
| ISO-C | 81.9 | 90.9 | 94.8 | 98.7 | 91.4 | 95.5 | 99.2 | 79.2 | 91.4 |
| ISO-CTS | 81.3 | 91.2 | 94.7 | 98.6 | 89.4 | 95.3 | 99.2 | 77.9 | 90.9 |
| AlignMerge-B | 82.2 | 90.7 | 94.9 | 98.6 | 93.7 | 96.8 | 99.4 | 81.5 | **92.2** |
| AlignMerge-O | 79.9 | 90.8 | 94.7 | 98.9 | 95.1 | 96.7 | 99.5 | 80.9 | 92.1 |

Table 7: Performance comparison across different methods and 8 datasets on ViT-L/14.

**Merging experiments on 14 CV benchmark** The performance of each method on individual datasets is presented in detail. Complete results for ViT-B/32, ViT-B/16, and ViT-L/14 are provided in Table 8, Table 9, and Table 10, respectively.

**Merging experiments on full fine-tuned BERT** The performance of each method on individual datasets is presented in detail. Complete results for BERT (Devlin et al., 2019) are provided in Table 11.

**Merging experiments on full fine-tuned T5** The performance of each method on individual datasets is presented in detail. Complete results for T5 (Raffel et al., 2020) are provided in Table 12.

**Merging experiments on PEFT fine-tuned T0** The detailed performance of each method on individual datasets is provided. IA3 (Liu et al., 2022) is employed for model Parameter-Efficient

| Method | Caltech101 | Cars | CIFAR100 | DTD | EuroSAT | FGVC | Flowers102 | Food101 | GTSRB | MNIST | OxfordPets | RESISC45 | SUN397 | SVHN | Avg. Acc. |
|---|---|---|---|---|---|---|---|---|---|---|---|---|---|---|---|
| Pretrained | 89.2 | 59.6 | 66.1 | 44.4 | 45.7 | 17.0 | 73.5 | 79.5 | 32.6 | 48.3 | 82.3 | 60.3 | 62.3 | 31.6 | 56.6 |
| Individual | 95.1 | 77.7 | 89.3 | 79.4 | 99.8 | 46.6 | 87.3 | 85.0 | 98.7 | 99.7 | 90.5 | 96.1 | 79.2 | 97.5 | 87.3 |
| TA | 91.4 | 62.4 | 72.0 | 46.8 | 64.8 | 18.6 | 73.5 | 78.9 | 44.0 | 76.9 | 85.1 | 67.0 | 63.5 | 51.6 | 64.0 |
| DARE | 91.6 | 62.4 | 71.7 | 46.7 | 64.3 | 18.5 | 73.5 | 78.9 | 43.7 | 76.7 | 85.1 | 67.0 | 63.6 | 51.4 | 63.9 |
| TIES | 86.6 | 55.9 | 69.7 | 47.1 | 70.6 | 30.0 | 49.0 | 65.1 | 53.0 | 87.7 | 69.8 | 63.5 | 59.0 | 55.1 | 61.6 |
| TSVM | 90.8 | 67.0 | 74.3 | 65.5 | 94.6 | 37.1 | 60.8 | 73.9 | 88.4 | 99.0 | 86.7 | 81.0 | 64.1 | 86.9 | 76.4 |
| ISO-C | 91.2 | 63.9 | 75.5 | 61.2 | 90.9 | 39.0 | 50.0 | 69.2 | 83.6 | 98.5 | 78.8 | 77.5 | 65.5 | 82.7 | 73.4 |
| ISO-CTS | 92.7 | 68.7 | 77.1 | 64.9 | 89.8 | 39.2 | 64.7 | 75.5 | 85.8 | 98.5 | 83.4 | 82.8 | 68.7 | 82.4 | 76.7 |
| AlignMerge-B | 91.8 | 71.0 | 81.2 | 66.3 | 92.2 | 38.1 | 61.8 | 77.7 | 81.8 | 97.5 | 85.1 | 84.1 | 70.7 | 75.7 | 76.8 |
| AlignMerge-O | 91.4 | 68.7 | 75.8 | 67.5 | 94.3 | 39.8 | 68.6 | 77.9 | 89.3 | 99.0 | 87.8 | 83.4 | 68.0 | 87.7 | 78.5 |

Table 8: Performance comparison across different methods and 14 datasets on ViT-B/32.

| Method | Caltech101 | Cars | CIFAR100 | DTD | EuroSAT | FGVC | Flowers102 | Food101 | GTSRB | MNIST | OxfordPets | RESISC45 | SUN397 | SVHN | Avg. Acc. |
|---|---|---|---|---|---|---|---|---|---|---|---|---|---|---|---|
| Pretrained | 86.7 | 64.7 | 69.6 | 44.7 | 54.6 | 25.1 | 67.7 | 85.7 | 43.4 | 51.7 | 87.2 | 66.4 | 63.8 | 52.0 | 61.7 |
| Individual | 97.4 | 86.8 | 89.7 | 82.0 | 99.8 | 54.7 | 82.4 | 90.3 | 99.1 | 99.8 | 94.6 | 96.8 | 81.7 | 97.8 | 89.5 |
| TA | 87.6 | 65.9 | 73.0 | 45.9 | 60.1 | 26.3 | 67.6 | 85.6 | 46.6 | 64.4 | 85.9 | 68.7 | 64.9 | 57.3 | 64.3 |
| DARE | 88.5 | 66.0 | 75.8 | 46.2 | 66.1 | 26.3 | 67.6 | 83.8 | 50.6 | 86.3 | 83.7 | 70.4 | 65.6 | 64.0 | 67.2 |
| TIES | 89.8 | 53.7 | 80.2 | 38.4 | 50.9 | 26.9 | 54.9 | 75.5 | 40.6 | 87.7 | 85.6 | 56.8 | 59.4 | 61.5 | 61.6 |
| TSVM | 90.2 | 69.9 | 77.9 | 54.7 | 91.6 | 35.3 | 66.7 | 81.5 | 85.7 | 98.8 | 92.9 | 80.2 | 64.4 | 88.5 | 77.0 |
| ISO-C | 94.4 | 51.7 | 79.5 | 46.1 | 85.1 | 35.4 | 57.8 | 75.9 | 79.0 | 98.4 | 92.9 | 73.3 | 58.8 | 84.3 | 72.3 |
| ISO-CTS | 93.7 | 71.3 | 77.0 | 58.8 | 93.8 | 39.6 | 74.5 | 80.9 | 90.8 | 98.9 | 93.8 | 81.4 | 65.8 | 89.3 | 79.2 |
| AlignMerge-B | 94.4 | 68.8 | 81.7 | 54.6 | 88.9 | 37.4 | 69.6 | 82.9 | 83.0 | 98.2 | 92.0 | 80.8 | 65.2 | 85.5 | 77.4 |
| AlignMerge-O | 90.6 | 73.6 | 78.2 | 59.2 | 93.0 | 39.2 | 72.5 | 83.3 | 88.0 | 98.8 | 94.0 | 83.5 | 66.9 | 89.0 | 79.3 |

Table 9: Performance comparison across different methods and 14 datasets on ViT-B/16.

| Method | Caltech101 | Cars | CIFAR100 | DTD | EuroSAT | FGVC | Flowers102 | Food101 | GTSRB | MNIST | OxfordPets | RESISC45 | SUN397 | SVHN | Avg. Acc. |
|---|---|---|---|---|---|---|---|---|---|---|---|---|---|---|---|
| Pretrained | 91.4 | 77.9 | 78.5 | 55.4 | 62.3 | 31.5 | 81.4 | 89.6 | 50.5 | 76.3 | 93.8 | 71.3 | 66.9 | 58.4 | 70.4 |
| Individual | 95.8 | 92.3 | 87.8 | 84.1 | 99.7 | 65.0 | 88.2 | 92.6 | 99.2 | 99.7 | 94.3 | 97.4 | 84.9 | 98.1 | 91.4 |
| TA | 91.8 | 78.9 | 80.0 | 56.8 | 68.7 | 32.4 | 82.4 | 89.7 | 54.7 | 84.2 | 94.0 | 74.3 | 67.7 | 64.3 | 72.9 |
| DARE | 91.8 | 79.8 | 82.0 | 59.5 | 81.4 | 32.3 | 84.3 | 89.6 | 62.4 | 93.2 | 94.6 | 78.6 | 69.4 | 71.8 | 76.5 |
| TIES | 93.5 | 78.3 | 79.1 | 61.4 | 88.0 | 30.9 | 83.3 | 85.4 | 73.2 | 98.1 | 93.8 | 83.6 | 71.4 | 81.8 | 78.7 |
| TSVM | 93.8 | 88.2 | 80.3 | 74.5 | 98.1 | 39.3 | 92.2 | 89.4 | 95.2 | 99.4 | 94.3 | 91.7 | 74.6 | 94.0 | 86.1 |
| ISO-C | 93.8 | 89.0 | 82.3 | 74.2 | 97.4 | 42.8 | 92.2 | 90.0 | 94.7 | 99.3 | 95.4 | 93.2 | 78.3 | 91.2 | 86.7 |
| ISO-CTS | 94.1 | 90.3 | 83.2 | 76.3 | 98.3 | 42.6 | 91.4 | 91.4 | 95.7 | 99.3 | 95.4 | 94.2 | 78.7 | 91.8 | 87.5 |
| AlignMerge-B | 93.8 | 89.1 | 83.6 | 73.8 | 97.0 | 42.5 | 92.2 | 90.3 | 93.1 | 99.1 | 95.1 | 94.0 | 78.3 | 89.5 | 86.5 |
| AlignMerge-O | 93.8 | 89.4 | 81.1 | 75.4 | 98.4 | 40.5 | 94.1 | 90.3 | 96.0 | 99.5 | 94.8 | 92.8 | 75.4 | 94.2 | 86.8 |

Table 10: Performance comparison across different methods and 14 datasets on ViT-L/14.

| Method | Ag_news | Rotten_Tomatoes | Cola | SMS | Avg. Acc. |
|---|---|---|---|---|---|
| TA | 94.2 | 51.0 | 33.8 | 89.1 | 67.0 |
| TIES | 88.7 | 47.0 | 41.5 | 79.9 | 64.3 |
| DARE | 90.8 | 50.8 | 33.9 | 87.5 | 65.8 |
| TSVM | 97.9 | 52.4 | 34.8 | 91.9 | 69.3 |
| ISO-C | 77.2 | 44.7 | 40.7 | 92.3 | 63.7 |
| ISO-CTS | 73.2 | 43.6 | 42.9 | 90.1 | 62.5 |
| AlignMerge-O | 97.4 | 53.4 | 36.4 | 92.1 | 69.8 |
| AlignMerge-B | 97.8 | 59.9 | 34.4 | 90.3 | **70.6** |

Table 11: Performance comparison across different methods and datasets on BERT.

| Method | rte | cb | winogrande | wic | wsc | copa | h-swag | story_cloze | anli-r1 | anli-r2 | anli-r3 | Avg. Acc. |
|---|---|---|---|---|---|---|---|---|---|---|---|---|
| TA | 40.6 | 53.1 | 34.4 | 60.9 | 37.5 | 48.4 | 21.9 | 50.0 | 34.4 | 40.6 | 34.4 | 41.5 |
| DARE | 40.6 | 53.1 | 31.2 | 59.4 | 37.5 | 50.0 | 21.9 | 50.0 | 34.4 | 40.6 | 34.4 | 41.2 |
| TIES | 40.6 | 56.2 | 34.4 | 71.9 | 37.5 | 68.8 | 21.9 | 59.4 | 34.4 | 43.8 | 31.2 | 45.5 |
| TSVM | 43.8 | 71.9 | 40.6 | 57.8 | 37.5 | 59.4 | 25.0 | 68.8 | 34.4 | 40.6 | 31.2 | 46.5 |
| ISO-C | 40.6 | 68.8 | 37.5 | 57.8 | 37.5 | 50.0 | 28.1 | 59.4 | 34.4 | 40.6 | 28.1 | 43.9 |
| ISO-CTS | 40.6 | 53.1 | 37.5 | 42.2 | 39.1 | 48.4 | 25.0 | 43.8 | 31.2 | 37.5 | 28.1 | 38.8 |
| AlignMerge-O | 42.2 | 75.0 | 50.0 | 51.6 | 37.5 | 56.2 | 28.1 | 65.6 | 34.4 | 40.6 | 31.2 | 46.6 |
| AlignMerge-B | 43.8 | 68.8 | 34.4 | 62.5 | 37.5 | 60.9 | 28.1 | 65.6 | 34.4 | 40.6 | 34.4 | 46.5 |

Table 12: Performance comparison across different methods and tasks on T5.

Fine-Tuning (PEFT). AlignMerge-B† indicates that the model is first merged using the TIES method, followed by the proposed subspace calibration approach. Experimental results demonstrate that AlignMerge-B† achieves significantly better outcomes, highlighting the advantage of jointly addressing interference through the integration of conflict and alignment. Complete results for T0 (Sanh et al., 2021) are provided in Table 13, Where methods rely on SVD (*e.g.*, TSVM, Iso-C, AlignMerge-O) are not applicable.

**Merging experiments on full fine-tuned Llama2** to evaluate the performance of the merged model on LLMs, the fine-tuned models WizardMath-7B-V1.0 (Luo et al., 2023) and Llama-2-7b-chat-hf (Touvron et al., 2023) are merged and tested on two benchmarks, AlpacaEval (Li et al., 2023) and GSM8K (Cobbe et al., 2021). Complete results for Llama2-7B (Touvron et al., 2023) are provided in Table 14.

| Method | rte | cb | winogrande | wic | wsc | copa | h-swag | story_cloze | anli-r1 | anli-r2 | anli-r3 | Avg. Acc. |
|---|---|---|---|---|---|---|---|---|---|---|---|---|
| TA | 89.1 | 62.5 | 53.1 | 51.6 | 50 | 87.5 | 50 | 93.8 | 56.2 | 46.9 | 43.8 | 62.2 |
| DARE | 89.1 | 62.5 | 56.2 | 53.1 | 50 | 89.1 | 50 | 93.8 | 56.2 | 46.9 | 43.8 | 62.8 |
| TIES | 84.4 | 84.4 | 53.1 | 56.2 | 53.1 | 84.4 | 43.8 | 90.6 | 59.4 | 53.1 | 53.1 | 65.1 |
| TSVM | – | – | – | – | – | – | – | – | – | – | – | – |
| Iso-c | – | – | – | – | – | – | – | – | – | – | – | – |
| Iso-CTS | – | – | – | – | – | – | – | – | – | – | – | – |
| AlignMerge-O | – | – | – | – | – | – | – | – | – | – | – | – |
| AlignMerge-B | 71.9 | 81.2 | 59.4 | 68.8 | 43.8 | 93.8 | 50 | 93.8 | 56.2 | 46.9 | 59.4 | 65.9 |
| AlignMerge-B† | 82.8 | 93.8 | 65.6 | 70.4 | 54.6 | 81.2 | 46.9 | 90.6 | 62.5 | 56.2 | 62.5 | 69.7 |

Table 13: Performance comparison across different methods and tasks on T0.

| Method | AlphcaEval (win_rate) | GSM8K (Acc. %) | Avg. |
|---|---|---|---|
| Task Arithmetic | 42.6 | 43.8 | 43.2 |
| TIES | 40.3 | 45.0 | 42.7 |
| DARE | 37.2 | 47.0 | 42.1 |
| TSV-M | 48.1 | 49.7 | 48.9 |
| ISO-C | 29.5 | 42.0 | 35.7 |
| ISO-CTS | 24.0 | 38.7 | 31.4 |
| AlignMerge_B (ours) | 48.3 | 47.9 | 48.1 |
| AlignMerge_O (ours) | 88.1 | 51.0 | 69.6 |

Table 14: Performance comparison across different methods on Llama2-7B.

## E.2 ADDITIONAL EXPERIMENTS

**Existence of alignment across different backbone.** In Fig. 1(b), we visualize subspace-alignment for ViT-B/32. To assess its prevalence, we additionally report ViT-L/14 and Llama2-7B in Fig. 6. For readability, we display only the top-32 singular subspaces. Across backbones, alignment is consistently concentrated in the top singular subspaces. Although the absolute scores for Llama2-7B seems are smaller, reflecting its larger matrix scale, the alignment pattern remains clear and pronounced.

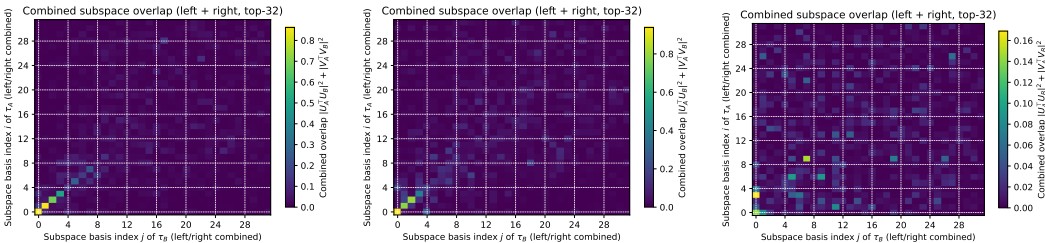

Figure 6: Visualisation of subspace-alignment (by subspace-overlap score) for the first linear layer of **ViT-B/16** (Left), **ViT-L/14** (Middle) and **Llama2-7B** (right).

**Statistical Analysis**. Since our method does not involve any randomness, we evaluate its effectiveness from a statistical perspective by constructing model merging tasks using random combinations of eight datasets from the main experiments, covering various numbers of models. We compare our AlignMerge-O method with TSVM in Fig. 9a, the results show that after conducting hundreds of experiments across different combinations, AlignMerge-O consistently outperforms the SOTA method.

**Order of Column and Row Projection.** In computing the Subspace Cross-Influence $\xi^r_{i,j}$, the paper adopts the default order of projecting onto the column subspace first, followed by projecting the residual onto the row subspace. However, this order is not mandatory. As shown in Fig. 7, we compute the coefficients in different singular vector subspaces of a random weight matrix. The results demonstrate that projecting onto the row subspace first and then onto the column subspace yields very similar coefficients.

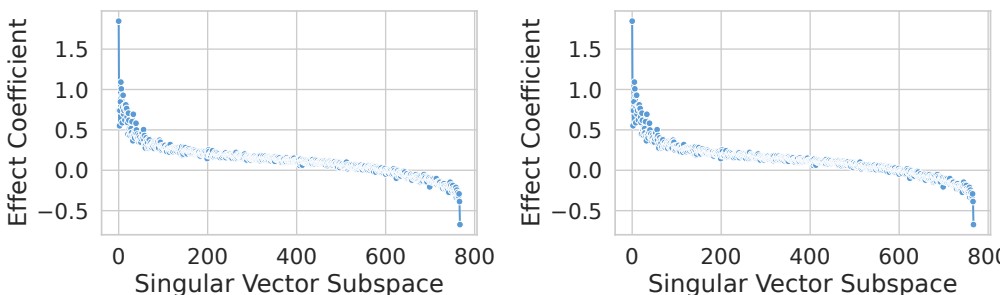

Figure 7: Visualisation of the Subspace Cross-Influence: the left shows projection onto the column subspace first, while the right shows projection onto the row subspace first.

**Experiments with Varying Numbers of Datasets.** As subspace-alignment are highly consistent across different models (shown in Fig. 6). Therefore, to demonstrate generalizability and reduce computational cost, for merging tasks involving more than 8 datasets, we use scaled coefficients derived from the eight-task scenario. Tab. 19 and Fig. 8 present results for fourteen- and twenty-task merging scenarios. Tab. 16 reports results for two-task merging, where eight tasks are randomly paired and the average accuracy across all pairs is indicated as "Avg Acc.". Fig. 9a further examines arbitrary combinations among the eight dataset models. These experiments collectively demonstrate that our method consistently yields robust improvements in varying scenarios.

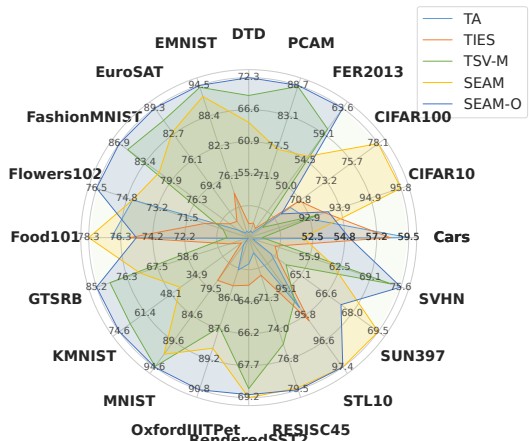

Figure 8: Merging performance on 20 tasks

**The Universality of AlignMerge-B.** In the main text, our AlignMerge-B method applies singular vector subspace correction after summing all weights. However, since subspace-alignment are prevalent in models produced by various merging methods, AlignMerge-B can also be effectively integrated into methods such as TIES (Yadav et al., 2023) and DARE (Yu et al., 2024). As shown in Tab. 15, we modify TIES-Merging by dividing each subspace of the merged model by its corresponding Subspace Cross-Influence, resulting in TIES w/ AlignMerge-B. The hyperparameters $\alpha$ are set the same as in the main paper. The results demonstrate that AlignMerge-B remains highly effective when applied to TIES, and the same holds for DARE w/ AlignMerge-B. With further tuning of these hyperparameters, performance is expected to improve even more.

Table 15: Applying AlignMerge-B on different merging methods.

| Method | SUN397. | Cars. | RESISC45. | EuroSAT. | SVHN. | GTSRB. | MNIST. | DTD. | Avg Acc |
|---|---|---|---|---|---|---|---|---|---|
| TIES | 59.6 | 58.59 | 70.96 | 81.33 | 86.14 | 70.85 | 98.36 | 54.73 | 72.57 |
| TIES w/ AlignMerge-B | 70.36 | 66.72 | 81.73 | 88.93 | 79.93 | 68.14 | 97.53 | 64.52 | 77.23 |
| DARE | 64.75 | 63.3 | 71.38 | 72.67 | 64.17 | 52.74 | 87.37 | 50.27 | 65.83 |
| DARE w/ AlignMerge-B | 64.91 | 65.07 | 79.75 | 93.04 | 89.92 | 84.77 | 99.08 | 61.28 | 79.73 |

**Robustness On Unseen Tasks.** Here, we evaluate zero-shot performance by merging models trained on 7 out of 8 datasets and testing on the remaining one. As shown in Fig. 9b, our AlignMerge-O consistently outperforms the SOTA TSV-M method in various scenarios.

**Alignment of Singular Vector Subspace.** As observed in the main text, the top singular vector subspaces of different task vectors exhibit high alignment. Here, we visualise the specific alignment between different singular vector subspaces. Specifically, we perform SVD on the task vectors of specialised models and extract the top-K singular vectors. By taking the outer product of the left and right singular vectors, we obtain a rank-1 matrix $\mathbf{M}_i$ for task $i$. We quantify the alignment between

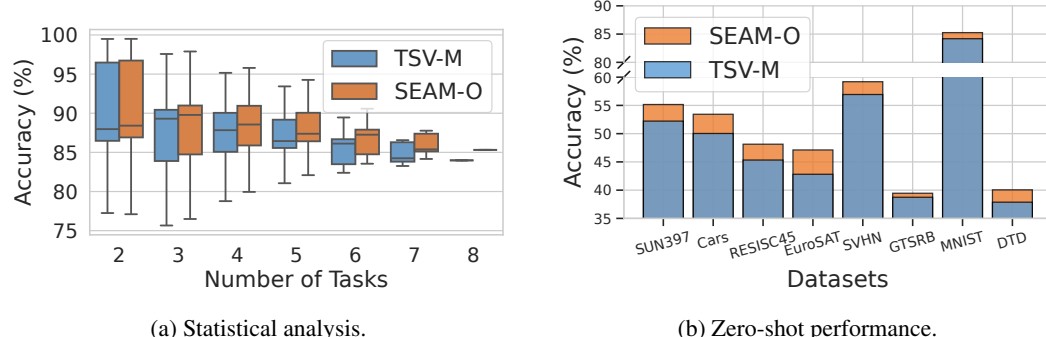

(a) Statistical analysis.

(b) Zero-shot performance.

Figure 9: Additional experiments (a) Statistical analysis based on arbitrary combinations of different models. (b) Evaluating the robustness on unseen tasks.

tasks $i$ and $j$ in the $K$-th singular vector subspace by computing $||\mathbf{M}_i^\top \mathbf{M}_j||_F$. Since both $\mathbf{M}_i$ and $\mathbf{M}_j$ are rank-1 matrices, $||\mathbf{M}_i^\top \mathbf{M}_j||_F = 1$ when they are identical, and $||\mathbf{M}_i^\top \mathbf{M}_j||_F = 0$ when they are orthogonal. The visualisation is shown in Fig. 10. Notably, the alignment between singular vector subspaces decreases from top-1 to top-4, and stabilises near zero beyond the top-4 subspace. This highlights the necessity of explicitly extracting the top-1 subspace in AlignMerge-O to prevent information loss due to orthogonalisation.

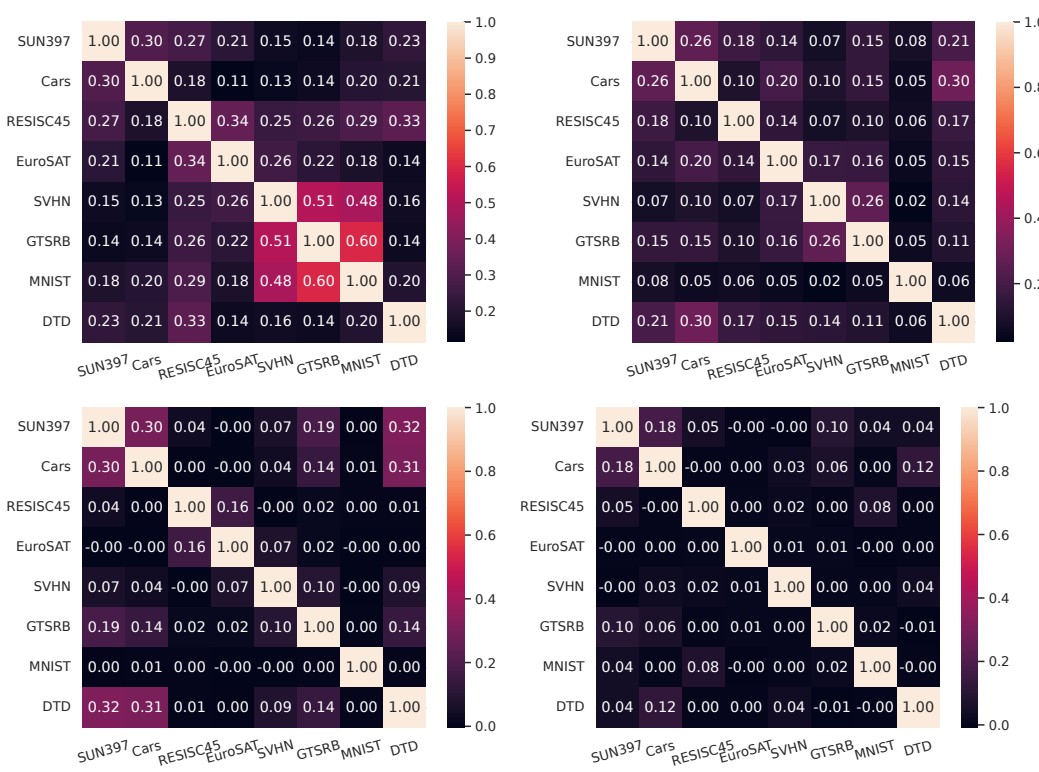

Figure 10: Visualisation of Singular Vector Subspace-Alignment. (Top left: Top-1 singular vector subspace; top right: Top-2; bottom left: Top-3; bottom right: Top-4)

**Effectiveness of $\eta$.** When modelling the relationship between the merged model and the specialized models, we introduce a scalar $\eta$ to ensure equality (Eq. 4). In AlignMerge-B, we define $\eta^r = \sigma^r_{\text{merge}} / \left( (\boldsymbol{u}^r_{\text{merge}})^\top \sum_{i=1}^{K} \Delta \mathbf{W}_i \boldsymbol{v}^r_{\text{merge}} \right)$. Whereas in AlignMerge-O, we define $\eta^r = \sigma^r_{\text{merge}} / \left( (\boldsymbol{u}^r_{\text{merge}})^\top \Delta \mathbf{W}_{i^*} \boldsymbol{v}^r_{\text{merge}} \right)$, ensuring the singular values on both sides of the equation are ex-

actly matched. Here, we visualise the singular values of the merged model obtained by AlignMerge-B, TIES w/ AlignMerge-B and AlignMerge-O, as well as the singular values obtained by directly projecting the specialised models onto the corresponding subspaces (Fig. 11). We observe that without $\eta$, the two singular values exhibit an approximately linear relationship. In AlignMerge-B, both singular values are exactly equal. In TIES w/ AlignMerge-B and AlignMerge-O, direct projection of the specialised models leads to singular values that are either excessively large or small compared to those of the merged model, respectively. Therefore, the absence of $\eta$ significantly increases the distance between the merged and specialized models in weight space.

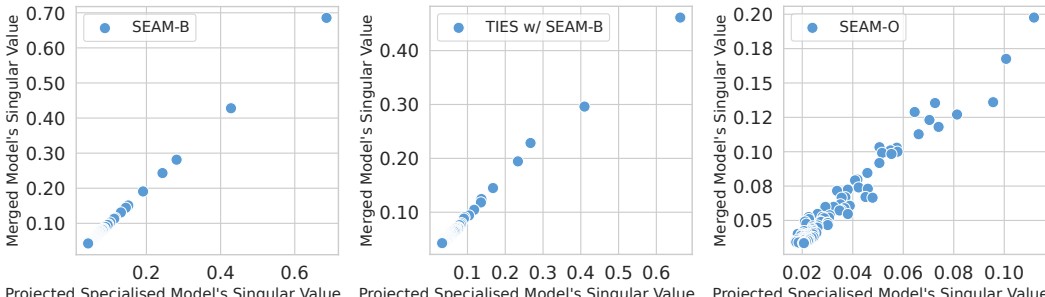

Figure 11: Visualisation of the singular values of the merged model compared to those of the projected specialised models.

Table 16: Multi-task performance (%) when merging ViT-B/32 models on random 2 task combinations. Combinations are: (EuroSAT, SUN397), (MNIST, DTD), (RESISC45, GTSRB), and (SVHN, Cars).

| Method | EuroSAT | SUN397 | MNIST | DTD | RESISC45 | GTSRB | SVHN | Cars | Avg Acc |
|---|---|---|---|---|---|---|---|---|---|
| TA | 98.8 | 73.8 | 99.5 | 69.3 | 91.1 | 95.3 | 95.4 | 72.0 | 86.9 |
| TIES | 98.7 | 75.1 | 99.6 | 71.1 | 92.8 | 95.6 | 96.3 | 73.8 | 87.9 |
| TSV-M | 99.8 | 76.6 | 99.7 | 76.8 | 94.4 | 98.4 | 97.3 | 74.8 | 89.7 |
| AlignMerge-B | 99.5 | 77.5 | 99.6 | 78.2 | 94.8 | 98.0 | 96.3 | 76.2 | 90.0 |
| AlignMerge-O | 99.5 | 77.5 | 99.6 | 77.1 | 95.1 | 98.2 | 96.9 | 76.7 | **90.1** |

## F    RELATIONSHIP TO RELATED WORKS

Current model merging methods can be broadly classified into static (Yu et al., 2024; He et al., 2024; Gargiulo et al., 2025) and dynamic methods (Huang et al., 2024; Yang et al., 2024b; Oh et al., 2024), with ours falling into the static category. The primary distinction lies in whether the model behaviour changes during inference for different inputs. In terms of space complexity, static methods typically require $\mathcal{O}(1)$ space, while dynamic methods require $\mathcal{O}(N)$, where $N$ is the number of tasks. Although dynamic merging methods generally offer better performance, they often require prior knowledge of the input dataset and cannot allow the model to perform multiple tasks simultaneously. As a result, they are unsuitable for important applications such as style mixture in image generation.

In addition, model merging methods can also be divided into training-required (Yang et al., 2024c;b) and training-free (Ilharco et al., 2022; Yadav et al., 2023) methods, with ours belonging to the training-free category. Training-required methods typically rely on large amounts of unlabeled data—often using unlabeled test samples—for unsupervised training via entropy minimisation, which incurs high computational and data costs. Nevertheless, some training-free methods also depend on additional data resources; for example, DaWin (Oh et al., 2024) requires substantial unlabeled data, while TA (Ilharco et al., 2022) and TIES (Yadav et al., 2023) require validation sets to tune scaling coefficients for task vectors. Therefore, in this work, we mainly compare with static methods that are easily applicable in practice and rely only on a validation set. Notably, our SEAM-O can operate entirely without any additional data.

Table 17: Multi-task performance (%) when merging ViT-B/32 models on eight tasks.

| Method | SUN397 | Cars | RESISC45 | EuroSAT | SVHN | GTSRB | MNIST | DTD | Avg Acc |
|---|---|---|---|---|---|---|---|---|---|
| Pretrained | 62.3 | 59.7 | 60.7 | 45.5 | 31.4 | 32.6 | 48.5 | 43.8 | 48.0 |
| Individual | 75.3 | 77.7 | 96.1 | 99.7 | 97.5 | 98.7 | 99.7 | 79.4 | 90.5 |
| Traditional MTL | 73.9 | 74.4 | 93.9 | 98.2 | 95.8 | 98.9 | 99.5 | 77.9 | 88.9 |
| Weight Averaging | 65.3 | 63.4 | 71.4 | 71.7 | 64.2 | 52.8 | 87.5 | 50.1 | 65.8 |
| TA | 55.2 | 54.9 | 66.7 | 78.9 | 80.2 | 69.7 | 97.3 | 50.4 | 69.1 |
| DARE | 64.9 | 63.5 | 71.8 | 72.4 | 63.8 | 52.4 | 87.5 | 50.5 | 65.8 |
| TIES | 59.8 | 58.6 | 70.7 | 79.7 | 86.2 | 72.1 | 98.3 | 54.2 | 72.4 |
| TSV-M | 69.1 | 70.7 | 85.5 | 94.3 | 92.0 | 91.9 | 99.3 | 69.2 | 84.0 |
| Iso-CTS | 74.4 | 74.4 | 87.2 | 90.4 | 76.8 | 83.3 | 97.4 | 67.0 | 81.4 |
| AlignMerge-B | 73.8 | 73.8 | 88.8 | 95.7 | 86.7 | 89.7 | 98.9 | 71.5 | 84.9 |
| AlignMerge-O | 72.0 | 74.2 | 86.9 | 94.7 | 91.0 | 92.7 | 99.2 | 71.8 | **85.3** |

Table 18: Multi-task performance (%) when merging models with different backbones on 8 tasks.

| Backbone | Method | SUN397 | Cars | RESISC45 | EuroSAT | SVHN | GTSRB | MNIST | DTD | Avg Acc |
|---|---|---|---|---|---|---|---|---|---|---|
| | TA | 61.1 | 65.9 | 74.0 | 76.2 | 88.0 | 73.9 | 98.4 | 53.0 | 73.8 |
| | TIES | 59.1 | 72.5 | 80.5 | 84.0 | 85.0 | 71.5 | 98.1 | 54.9 | 77.0 |
| ViT-B/16 | TSV-M | 72.8 | 80.4 | 89.1 | 96.6 | 93.9 | 93.9 | 99.3 | 72.7 | 87.3 |
| | AlignMerge-B | 76.3 | 81 | 91.7 | 97.6 | 92.1 | 94.2 | 99.1 | 75.8 | 88.5 |
| | AlignMerge-O | 75.4 | 82.9 | 90.7 | 98.0 | 93.0 | 95.0 | 99.2 | 75.6 | **88.8** |
| | TA | 74.0 | 81.9 | 86.5 | 92.9 | 87.8 | 86.8 | 99.0 | 66.1 | 84.4 |
| | TIES | 74.4 | 84.5 | 88.0 | 94.0 | 85.7 | 82.1 | 98.7 | 67.7 | 84.4 |
| ViT-L/14 | TSV-M | 79.1 | 89.9 | 94.0 | 98.9 | 95.3 | 96.2 | 99.5 | 79.1 | 91.5 |
| | AlignMerge-B | 82.2 | 90.7 | 94.9 | 98.6 | 93.7 | 96.8 | 99.4 | 81.5 | **92.2** |
| | AlignMerge-O | 79.9 | 90.8 | 94.7 | 98.9 | 95.1 | 96.7 | 99.5 | 80.9 | 92.1 |

Table 19: Multi-task performance (%) when merging ViT-B/32 models on 14 tasks.

| Method | Caltech101 | Cars | CIFAR100 | DTD | EuroSAT | FGVC | Flowers102 | Food101 | GTSRB | MNIST | OxfordPets | RESISC45 | SUN397 | SVHN | Avg Acc. |
|---|---|---|---|---|---|---|---|---|---|---|---|---|---|---|---|
| TA | 91.4 | 62.4 | 72 | 46.8 | 64.8 | 18.6 | 73.5 | 78.9 | 44 | 76.9 | 85.1 | 67 | 63.5 | 51.6 | 64.0 |
| TIES | 86.6 | 55.9 | 69.7 | 47.1 | 70.6 | 30.0 | 49.0 | 65.1 | 53.0 | 87.7 | 69.8 | 63.5 | 59.0 | 55.1 | 61.6 |
| TSV-M | 90.8 | 67.0 | 74.3 | 65.5 | 94.6 | 37.1 | 60.8 | 73.9 | 88.4 | 99.0 | 86.7 | 81.0 | 64.1 | 86.9 | 76.4 |
| AlignMerge-B | 91.8 | 71.0 | 81.2 | 66.3 | 92.2 | 38.1 | 61.8 | 77.7 | 81.8 | 97.5 | 85.1 | 84.2 | 70.7 | 75.7 | 76.8 |
| AlignMerge-O | 91.4 | 68.7 | 75.8 | 67.5 | 94.3 | 39.8 | 68.6 | 77.9 | 89.3 | 99.0 | 87.8 | 83.4 | 68.0 | 87.7 | **78.5** |

# G DETAILED VISUALISATION

We paired models trained on each of the eight datasets in our main experiments and visualised the subspace-alignment across different layers for each pair. Experimental results are presented in Figs. 12, 13, 14, 15, and 16. The results indicate that while effect coefficients exhibit similar overall patterns across different pairs, notable differences remain; moreover, subspace-alignment generally dominate over conflict effects.

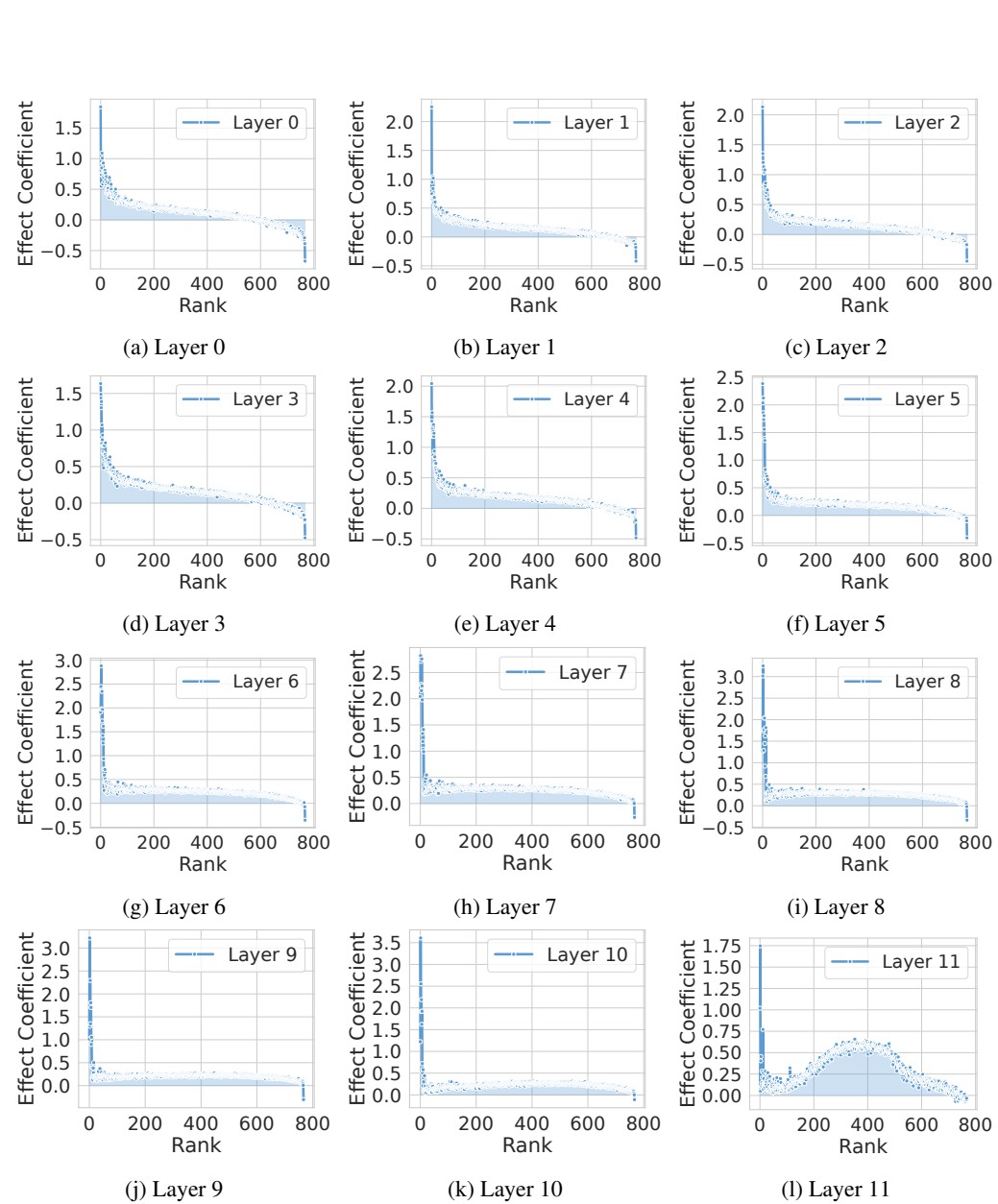

Figure 12: Subspace Cross-Influence that "MNIST" impact on "Cars" for all layers.

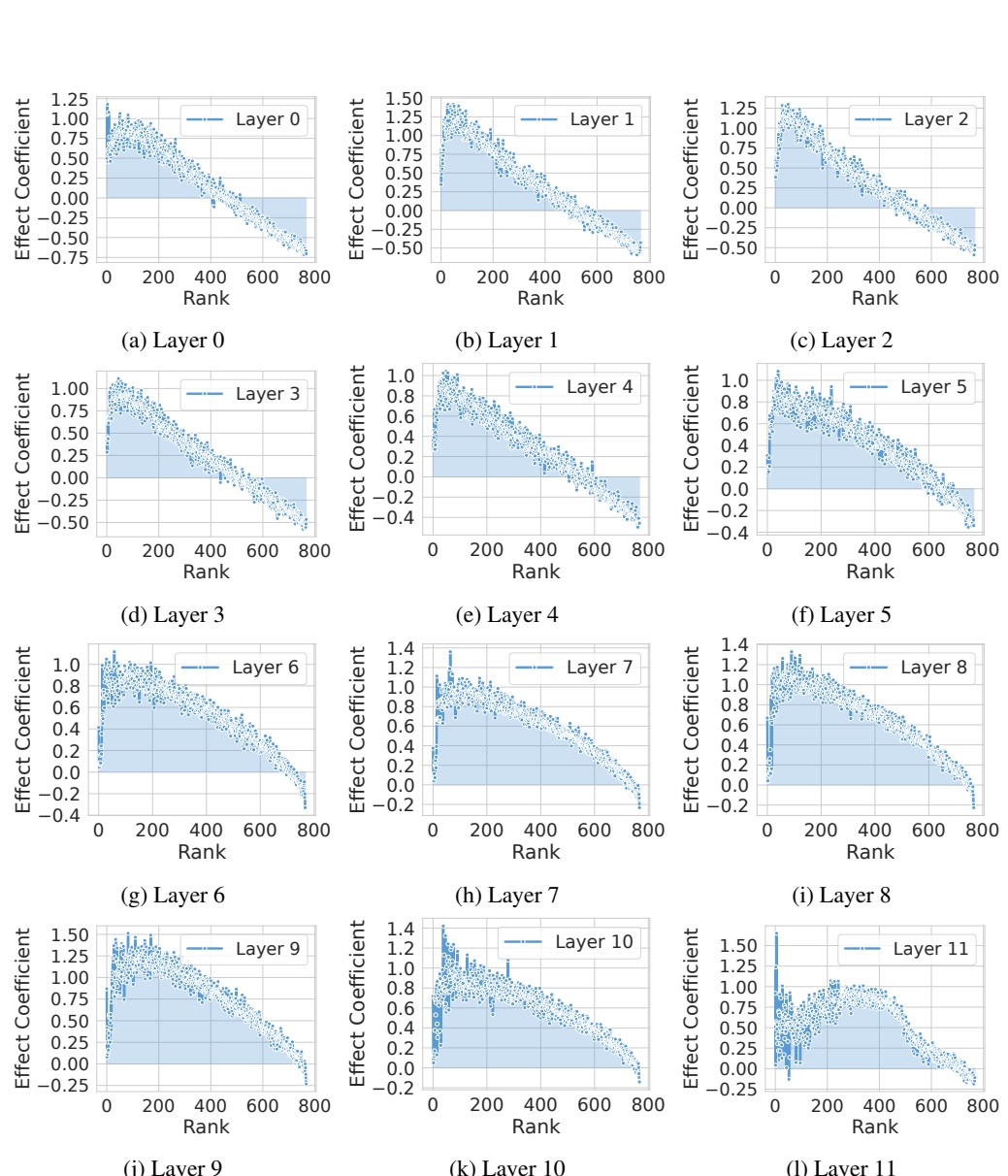

(a) Layer 0        (b) Layer 1        (c) Layer 2

(d) Layer 3        (e) Layer 4        (f) Layer 5

(g) Layer 6        (h) Layer 7        (i) Layer 8

(j) Layer 9        (k) Layer 10        (l) Layer 11

Figure 13: Subspace Cross-Influence that "Cars" impact on "MNIST" for all layers.

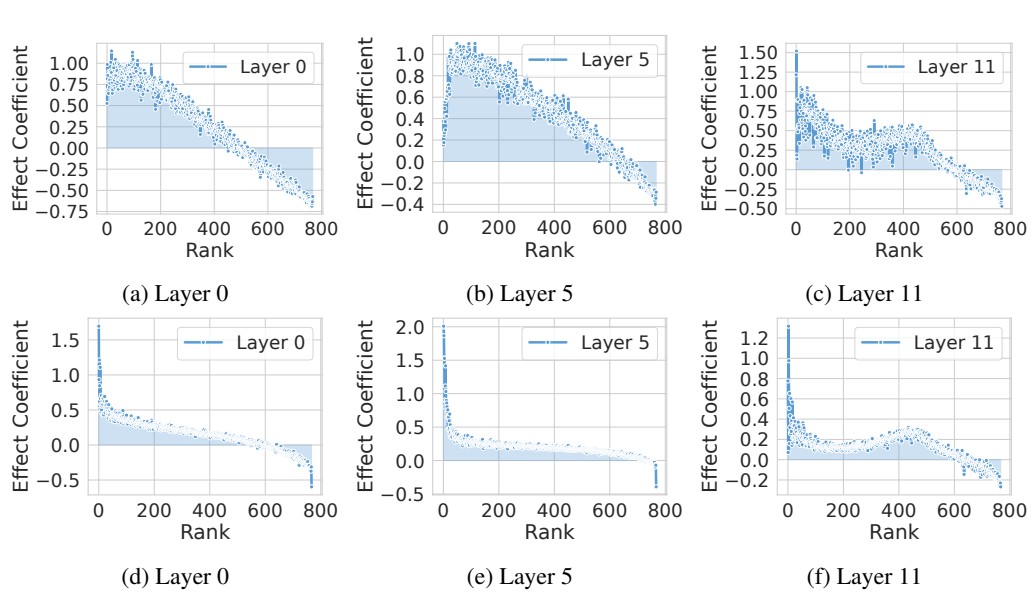

Figure 14: Subspace Cross-Influence that "SUN397" impact on "GTSRB" at first row and "GTSRB" impact on "SUN397" at second row.

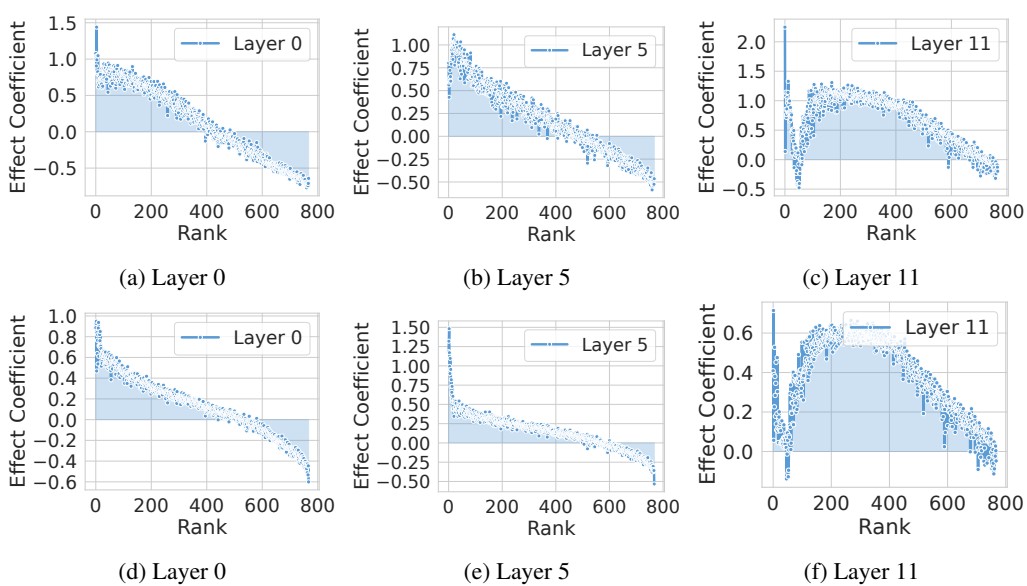

Figure 15: Subspace Cross-Influence that "RESISC45" impact on "EuroSAT" at first row and "EuroSAT" impact on "RESISC45" at second row.

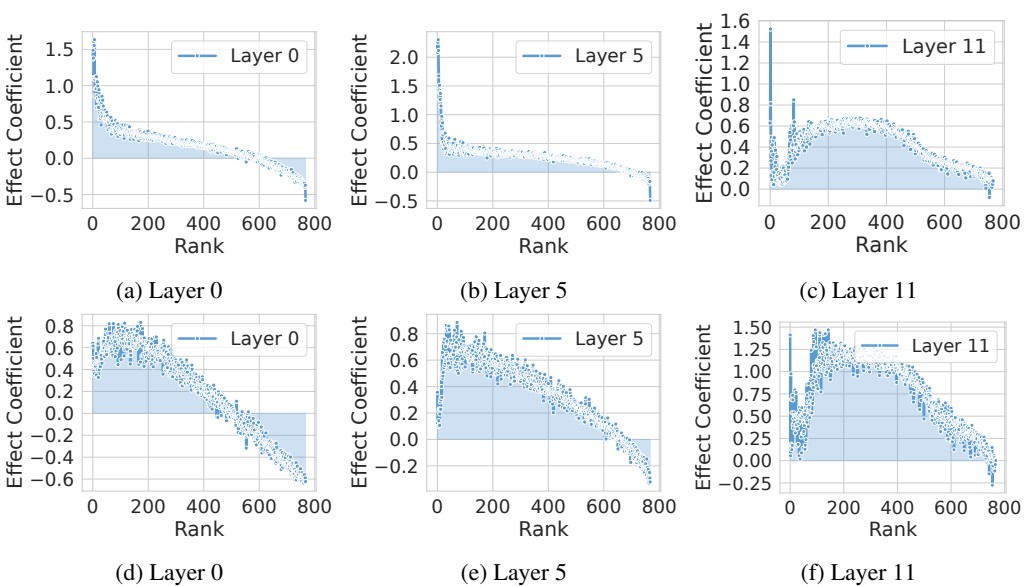

Figure 16: Subspace Cross-Influence that "SVHN" impact on "DTD" at first row and "DTD" impact on "SVHN" at second row.

