# OpenReview forum: "Beyond Conflict: Subspace-Alignment as the Missing Piece of Model Merging"
_ICLR.cc/2026/Conference — Submitted to ICLR 2026_

### Official Review · Reviewer_HBD5 · 2025-10-28

**Soundness:** 2
**Presentation:** 3
**Contribution:** 2
**Rating:** 4
**Confidence:** 4

**Summary:**

Paper proposes a new method for model merging. It argues that previous work has focused on preventing drift, and this paper instead analysis conflict from the view of alignment between task vectors. This can lead to spectral imbalance and the method proposes to address this problem. The paper follows recent works on SVD based mering (Gargiulo 2025, Marczak 2025).  Results show good results on vision and language model merging tasks.

**Strengths:**

- The authors might have come up with  a better explanation for the functioning of SVD based model merging methods (although they do not claim their explanation could be applied to previous methods, I think that might be true). The alignment story provides a good motivation why the SVD methods actually improve model merging.
- ablation of the method is good showing the impact of the several factors on final performance.

**Weaknesses:**

- The paper is building on recent works of SVD based model merging (these methods saw a significant jump with respect to TA). The authors do a poor job in explaining the main difference of their method with respect to these methods (much of their alignment story could be hold for these methods ? ). It is the task of the authors to clearly distinguish from the most relevant related work, and the authors have not done that.
- the results in Table one for TSV-M, Iso_C, Iso-CTS are significantly lower than in original papers (often outperforming the proposed method). The authors should be clear about the reason for this difference (beyond different checkpoints); any bugs in ISO-C ?
- Figure 1 too complex, not clear for me. Needs improvement.

**Questions:**

I would really want the authors to do a much more detailed discussion of the differences of their method with the previous SVD based methods. Alignment plays an important role in the motivation of Marczak as well. It is not the task of the reviewer to find the differences, but the authors should provide that.

Please address the drop in performance of ISO-C in tables in more detail.

---

> ### Author Response · Authors · 2025-11-23
> **Response to Weakness 1&2**
>
> We sincerely appreciate the reviewers' valuable feedback and are grateful for the opportunity to address these comments.
>
> **W1: Difference with Other SVD Based Methods**
>
> The manuscript currently lacks detailed comparisons with other SVD-based methods and relevant content will be added to the related work section. The primary comparisons considered include TSV-M and Iso-C here.
>
> TSV-M identifies interference among subspaces associated with different tasks. The concept of interference in TSV-M corresponds to the Alignment proposed in this work. If the singular vectors are strictly orthogonal, and thus the respective subspaces are orthogonal, interference does not occur. TSV-M mitigates interference by orthogonalizing the subspaces. In contrast, in our paper, we think that **interference implies potential alignment**, which further indicates the possibility of leveraging shared knowledge. Therefore, our AlignMerge-O extracts shared knowledge from the top subspace as defined in Eq. (11).
>
> Iso-C and our approach differ significantly in the motivations. Both propose subspace alignment, although the definition of subspace alignment in Iso-C is distinct from that in our work. Iso-C focuses on alignment between the specialized model and the merged model subspaces, aiming to enhance alignment between specialized models and the merged model. However, this alignment is primarily superficial; effective alignment between specialized models and the merged model presupposes alignment among specialized models themselves. Interestingly, less alignment (*i.e.*, greater orthogonality) among specialized models yields stronger alignment between each specialized and the merged model. Consequently, **our work emphasizes subspace alignment among specialized models** themselves.
>
> **W2: Different Performance**
>
> It is evident that the observed performance degradation primarily **arises from differences in checkpoints and the intrinsic lack of robustness of the Iso-C method**.
>
> On one hand, varying checkpoints imply that individual specialized models exhibit differing performance upper bounds on the corresponding tasks. Furthermore, empirical observations during finetuning indicate that modifications in training epochs and learning rates can substantially affect both the difficulty and outcomes of checkpoint merging. **Notably, despite these variations, TSV-M consistently demonstrates significant improvements over other merging methods, while Iso-C can not.** The reproduced experiments in this work employ checkpoints provided by classic Task Arithmetic, and the implementation is directly copied from the official [repo](https://github.com/danielm1405/iso-merging/blob/main/src/utils/iso.py), thereby minimizing the likelihood of reproduction errors.
>
> On the other hand, the Iso-C method itself exhibits structural instability. We conduct experiments on the original checkpoints associated with the Iso-C paper, which reveal that, while SOTA performance is achieved with the merging of 8 or more CV models, **a significant performance drop occurs when merging any 2 models**, sometimes even worser than basic Task Arithmetic method (it's welcome for the reviewers to reproduce). This phenomenon can be explained by perturbation theory [1], which suggests that subspaces corresponding to small singular values are predominantly occupied by noise. When only a few models are merged, these subspaces in the merged model still remain heavily contaminated by noise. Scaling up their singular values to the average level inevitably degrades model performance. As the number of merged models increases, meaningful components progressively replace the noisy elements in these bottom subspaces.
>
> A straightforward remedy for Iso-C involves rescaling only a small subset of top singular values to their mean while keep other unchanged; this modification ensures more robust merging performance even in cases involving the merger of just two CV models.
>
> [1]. Perturbation theory for the singular value decomposition.

---

> > ### Author Response · Authors · 2025-11-23
> > **Response to Weakness 3**
> >
> > **W3: Clarity of Figure 1**
> >
> > We apologize for any confusion caused. In fact, the core of Figure 1 lies in part (a). The intention is to convey that although task vectors for different tasks tend towards orthogonality, **their matrix representations, particularly subspaces obtained through SVD, show substantial overlap (i.e., alignment).** Notably, subspaces corresponding to high singular values across different tasks exhibit pronounced alignment, which can be quantitatively assessed using the cosine similarity of singular vectors. In contrast, subspaces associated with smaller singular values demonstrate minimal overlap.
> >
> > It is well understood that for vector analogies, higher alignment results in a greater norm after summation (amplify corresponding singular value in the merged model), whereas orthogonal vectors yield a smaller norm upon summation. This observation motivates an adaptive adjustment of different subspaces based on their degree of alignment. For instance, explicitly reducing the singular values of highly aligned subspaces can prevent over-amplification of activations associated with these subspaces, thereby mitigating adverse effects on performance.
> >
> > This perspective also accounts for the effectiveness of Iso-C, as averaging singular values in the spectral space suppresses highly overlapping subspaces by reducing large singular values. However, **Iso-C adopts a coarse approach and does not provide justification for why global averaging is optimal**, which in turn limits its performance relative to our method.
> >
> > Future work will include further refinement of the illustrations and soliciting experts from other domain to enhance the overall readability.

---

### Official Review · Reviewer_Pmir · 2025-10-29

**Soundness:** 2
**Presentation:** 3
**Contribution:** 2
**Rating:** 4
**Confidence:** 4

**Summary:**

This paper proposes and analyzes subspace alignment as a key but overlooked factor causing interference in model merging. The authors observe that although task vectors are approximately orthogonal overall, they exhibit significant directional alignment in the top-level singular subspaces sorted by singular values. During merging, the singular values ​​of these aligned subspaces accumulate and amplify, leading to over-activation of these subspaces during inference and thus degrading overall performance. Based on this, the authors propose two training-free methods: AlignMerge-B (which calculates calibration coefficients per subspace after merging and scales/suppresses over-aligned components); and AlignMerge-O (which extracts and preserves the "aligned subspaces" before merging while orthogonalizing/whitening the white noise/remaining subspaces to avoid cumulative amplification). The paper shows that AlignMerge achieves or approaches state-of-the-art performance on multiple benchmarks and narrows the gap with traditional multi-task learning to approximately 3.6%.

**Strengths:**

1. The writing is quite easy to read and it was well-written
2. The gains from their proposed method of sampling are convincing and comprehensive
3. The article shifts the focus of model fusion from "conflict" to "subspace alignment," and points out that the alignment of top-level singular subspaces will produce a cumulative amplification of singular values/activations after merging, filling a major blind spot in the existing explanation.

**Weaknesses:**

1. The paper mentions that "the top subspace alignment is significant" and that the top-1 is treated specially, but it does not provide a more detailed explanation of how to select which subspaces in different layers or modules, how to select the number of singular values, and the application of this selection to each task.
2. The authors primarily treat "alignment" as a new factor besides conflict, but do not fully explore how to combine conflict-mitigation with AlignMerge to achieve complementary gains. I am curious whether merging the two can further narrow the gap with MTL.

**Questions:**

1.  Theories on orthogonalizing task vectors have been proposed in AWD[1]. Whether there is a fundamental difference between orthogonalizing in the singular value space after SVD and directly orthogonalizing task vectors is something that needs further explanation in this paper.
2. The methods compared in the article are limited. I would like to know whether the method proposed in the article has any further advantages compared to some recent SOTA studies, such as [2][3][4].
3. The article states that SVD only applies to some linear layers. Could you please specify which layers were selected and the selection criteria?

[1] Multi-task model merging via adaptive weight disentanglement

[2] EMR-Merging: Tuning-Free High-Performance Model Merging

[3] Whoever Started the Interference Should End It: Guiding Data-Free Model Merging via Task Vectors

[4] Task Singular Vectors: Reducing Task Interference in Model Merging

If the author can solve the question and the weakness well, i will raise my score.

---

> ### Author Response · Authors · 2025-11-23
> **Response**
>
> We appreciate the reviewers’ thorough evaluation and are pleased to address the comments and suggestions provided.
>
> **W1: Methodology of Subspace Selection**
>
> Apologies for any confusion. The direct reference to “the top subspace alignment is significant” cannot be found, as the manuscript does not include any explicit subspace selection procedure. The only relevant content may appears in Section 4.2, where AlignMerge-O selects the top-1 subspace from each task-specific matrix to extract shared subspace. Specifically, for each model, the **top-1 subspace of each layer is defined as the space spanned by the left and right singular vectors corresponding to the largest singular value**, obtained via SVD of the layer weights. By default, only the first singular value is selected for each model, so the number of selected singular values is one. No other subspace selection method is implemented, as the procedure simply chooses the largest singular value in each layer for each task’s model, as shown in Eq. (11). Further clarification is welcome if this interpretation does not address the question. In conclusion, the method selects only the top-1 singular value/subspace per layer, without using an explicit selection procedure.
>
> **W2: Combination with Conflict-Mitigation Methods**
>
> It is also believed that integrating conflict mitigation with AlignMerge can yield improved performance. However, due to the already considerable length and scope of the current paper, further expansions may significantly diminish its readability. Therefore, this topic is reserved for future investigation.
>
> **Q1: Orthogonalization Difference**
>
> The orthogonality in singular value space is not the primary focus of this paper, it merely represents one specific implementation of AlignMerge-O, which directly adopts the approach of TSV-M. Therefore, detailed discussion is omitted. It can be further supplemented that, mathematically, **weight matrices inherently encapsulate the properties of their vectorized forms**. Specifically, when two matrices $\mathbf{A}$ and $\mathbf{B}$ of the same shape are orthogonal, i.e., ($\mathbf{A} \mathbf{B}^T = \mathbf{0}$), this implies ($\mathbf{0} = \text{trace}(\mathbf{A} \mathbf{B}^T) = \text{trace}(\mathbf{B}^T \mathbf{A}) = \langle \mathrm{vec}(\mathbf{B}), \mathrm{vec}(\mathbf{A}) \rangle$), indicating that their vectorizations are orthogonal. However, the converse does not hold. As a result, requiring orthogonality in the vectorized form is not meaningful when the weights are inherently represented as matrices.
>
> **Q2:  Comparison With Other Methods**
>
> First, our study have **provided a comprehensive comparison with [4]**, which corresponds to the repeatedly mentioned TSV-M method. However, there may have been inadvertent miscitation during manuscript preparation, attributing TSV-M to another study; this issue will be addressed.
>
> Secondly, **[2] relies on an extra task-specific mask matrix** that must be loaded according to the task type during inference. Similar to several mixture-of-experts (MoE) based methods, it represents a dynamic merging approach. Therefore, a direct comparison with our static methods is inappropriate. As a result, most related literature, including [4], does not include such comparisons.
>
> Finally, **[3] depends on additional unsupervised training**, potentially leading to increased training overhead. Although its employed checkpoint is nearly identical to that adopted in our study, [3] reports substantially higher accuracy on the SUN397 dataset (75.3% to 79.2%), which may be attributed to further fine-tuning of the checkpoint on SUN397. Compared to the our approach, [3] utilizes a stronger checkpoint and introduces additional training overhead. Nevertheless, across eight classic image classification tasks, our training-free method still achieves an accuracy of 85.3%, surpassing the training-required accuracy of 85.2% reported in [3].
>
> We clarify comparisons: TSV-M ([4]) is covered, [2] is dynamic/inappropriate for static comparison, and our training-free method outperforms the training-required method ([3]) on 8 tasks.
>
> **Q3: Layers Selection**
>
> Thank you for your valuable feedback. Some confusion may have arisen due to wording. The intended expression is that linear layers represent only a subset of all layers. In practice, consistent with prior SVD-based approaches (including TSV-M and Iso-C), all linear layers are processed accordingly, while the remaining layers are handled via weight averaging. **No deliberate selection or selection criteria are involved in this process.**

---

> ### Comment · Reviewer_Pmir · 2025-11-26
> **Reply to author's rebuttal**
>
> I thank the authors for answering my questions. I still have some concerns.
> 1. **Combination with Conflict-Mitigation Methods**:  I still believe that exploring the relationship between "alignment" and conflict resolution is an indispensable part of this article's exploration. This is because the author defines alignment as a new factor, but whether it is essentially another form of conflict resolution remains unknown. Furthermore, since the author did not conduct experiments combining the two, I remain skeptical of this claim.
> 2. **Comparison With Other Methods**:  Since the authors claim that other methods utilize additional resources, they should provide supplementary experiments to prove this. For example, the authors mention that [1] requires unsupervised training, but as can be seen from [1], the computational resources consumed are very small. Therefore, simply not requiring unsupervised training is not the advantage of the method. The authors should provide specific experimental data such as computation time, accuracy, and memory consumption to support their claim.
>
> For these reasons, I maintaing my reject score.
>
> [1] Whoever Started the Interference Should End It: Guiding Data-Free Model Merging via Task Vectors

---

### Official Review · Reviewer_2pcv · 2025-10-29

**Soundness:** 2
**Presentation:** 1
**Contribution:** 1
**Rating:** 2
**Confidence:** 4

**Summary:**

The paper deals with the problem of Model Merging, which consists of merging multiple task vectors obtained from multiple fine-tuned models on each task to obtain a final multi-task task vector. The authors argue that Subspace Alignment is another factor, beyond conflict, contributing to the interference among task vectors — a factor overlooked by previous model merging methods. Hence, they analyze the alignment between the task vectors and notice that top singular subspaces are strongly aligned, and that this alignment causes performance degradation and should therefore be removed. The authors propose a metric called Subspace Cross-Influence ($\upxi^{r}$), measuring the alignment between two tasks, and define the Subspace Alignment Index (SAI), measuring the interference as the deviation of $\upxi^{r}$ from its optimal value of one. The authors propose two complementary strategies, called Subspace-Alignment Aware Merging Through Balancing (AlignMerge-B) and Subspace-Alignment Aware Merging Through Orthogonalization (AlignMerge-O), which remove alignment in the post-merging and pre-merging phases, respectively. The method is evaluated across vision and language tasks.

**Strengths:**

- By removing the subspace alignment between task vectors the approach reduces over-amplication improving the performance.
- The approach provides  good performance with respect to the baselines.

**Weaknesses:**

1) The papers suffers from serious presentation issues that make it difficult to assess the soundness of the derivations in Section 3.2 and 3.3, which are necessary for understanding the methodologies:
- The notations used throughout the paper are heavy and significantly reduces  readability. The mathematical formulations lack intuitive explanations — especially geometric interpretations –  and many symbols are not clearly defined. For instance, the expression $$Proj_{P^r_{col}\,\Delta W_i}\left(P^r_{col}\,\Delta W_j\right) $$  is not properly introduced or explained in the text, and it represents an example of excessively heavy notation. Moreover, the use of  bold font  for all the  mathematical symbols  makes most of the formulas even harder to read.

- The paper is not self-contained. Understanding key components of the paper requires consulting the appendix, which should not be necessary.
For example, at Line 210, the Subspace Cross-Influence index (one of the main contributions) involves a term $W_{\text{res}}$​, whose definition appears only in the appendix. Since this index is crucial to understanding the proposed method, it should be clearly defined in the main text. The appendix should serve to provide clarifications or supplementary material – not essential definitions.

- The figures are difficult to interpret. In particular, Figure 1 is extremely intricate, and it is unclear how the various components are related. A clearer, more structured visual explanation would greatly improve comprehensibility.

2) The contribution seems largely incremental over [1], with several overlapping ideas and methodologies:

- The  Subspace Alignment as an overlooked  source of interference (the first contribution emphasized in the title and mentioned at Line 98 in the introduction), has already been introduced in [1]. The authors are aware of  this work,  as they cite and compare it in the experimental section, however this reference should appear in several points of the paper. For instance,  the discussion of Subspace Alignment as a motivation of interference (Section 3.2) is not novel. Many of the arguments presented between lines 180 to 190 (e.g. orthogonal task vectors and alignment highlights) have already been discussed in [1].

- Moreover,  [1]  introduced the concept of *Subspace Alignment Ratio* (SAR) to quantify subspace alignment, while the present paper proposes Subspace Cross-Influence and  *Subspace Alignment Index* (SAI). The authors should directly compare their metrics to SAR and clarify the differences.

-  (Line 324) Selecting the top-$k$ singular vectors to identify the most aligned, shared subspace is similar to the approach used in Iso-CTS [1], although in Iso-CTS this is computed starting from Task Arithmetic rather than from each task-specific matrix

- (Line 329) Averaging the singular values (Eq. 11) is the same operation performed in Iso-C and Iso-CTS [1].

- (Line 337) Performing orthogonal projection to obtain the residual (Eq. 12) is also done in Iso-CTS [1].

- (Line 347) Using whitening to obtain orthogonal bases is employed as well in both TSV-M [2] and Iso-CTS [1]

The authors must clearly justify why their proposed metrics are superior to those in [1], and explain why their method achieves performance improvements over [1], as reported in the experimental section.

[1] Marczak, D., Magistri, S., Cygert, S., Twardowski, B., Bagdanov, A.D. &amp; Van De Weijer, J.. (2025). No Task Left Behind: Isotropic Model Merging with Common and Task-Specific Subspaces. Proceedings of the 42nd International Conference on Machine Learning.

[2] Gargiulo, A. A., Crisostomi, D., Bucarelli, M. S., Scardapane, S., Silvestri, F., & Rodolà, E. (2025). Task singular vectors: Reducing task interference in model merging. In Proceedings of the IEEE/CVF Conference on Computer Vision and Pattern Recognition (CVPR).

**Questions:**

The paper requires substantial reformatting to improve clarity in notation, mathematical presentation, and overall readability. It would also benefit from the inclusion of intuitive geometric figures to illustrate the geometrical operations the authors apply to the task matrices. In its current form, the manuscript is not ready for publication. Furthermore, the narrative should be significantly revised. Since Subspace Alignment is a central concept of the work—one that has already been introduced in [1]—the preliminary section should discuss [1] in detail. Throughout the manuscript, the authors should clearly articulate how their approach and findings extend or improve upon [1], and explain why the proposed method achieves better performance.

---

> ### Author Response · Authors · 2025-11-23
> **Response to Weakness 1**
>
> Thank you for the detailed and insightful feedback. We have carefully considered each point and provide our responses below.
>
> After thoroughly reviewing the comments, it appears that the distinction between our AlignMerge and Iso-C has not been clearly understood. **In terms of motivation**, the two approaches originate from fundamentally different perspectives: Iso-C focuses on the alignment between specialized models and the merged model, whereas our AlignMerge examines how the alignment among specialized models themselves affects the performance of the merged model. **In terms of methodology**,  Iso-C and Iso-CTS employ a coarse singular values averaging strategy, which is demonstrated to be highly unstable. For instance, in two-model merging scenarios, the Iso-C method frequently fails, often resulting in performance even lower than that of classic task arithmetic method. **According to perturbation theory [1], the tail singular vectors are typically dominated by noise. The averaging operation in Iso-C amplifies the tail singular values, which essentially magnifies this noise and consequently leads to degraded model performance.** In contrast, our AlignMerge-B adaptively adjusts the singular values by analyzing the alignment of multiple task matrices within different subspaces, thereby achieving superior and more robust performance.
>
> Below is our point to point response.
>
> **W1: Presentation Issues**
>
>
> * **W1.1 Notation and Readability**
>
>   Efforts are made to simplify the formulas; however, the inclusion of certain notations is necessary to maintain mathematical rigor. For instance, the `Proj` expression you mentioned here is explicitly defined in Eq. (3) as $\mathrm{Proj}_\mathbf{B}(\mathbf{A})=\frac{<\mathbf{A}, \mathbf{B}>_F}{<\mathbf{B}, \mathbf{B}>_F}$.
>
>   Furthermore, the use of bold font is widely regarded as a more rigorous writing convention. In the preparation of the manuscript, **matrices** are represented by bold uppercase letters, **vectors** by bold lowercase letters, and **scalars** by italic lowercase letters.
>
> * **W1.2 Paper Self-Containment**
>
>   This feedback is highly valuable for us. Currently, the relevant content is included in the appendix primarily due to limitations in article length; this aspect will be further revised in subsequent versions.
>
> * **W1.3 Figure Interpretation**
>
>   We apologize for any confusion caused. In fact, the core of Figure 1 lies in part (a). The intention is to convey that although task vectors for different tasks tend towards orthogonality, **their matrix representations, particularly subspaces obtained through SVD, show substantial overlap (i.e., alignment).** Notably, subspaces corresponding to high singular values across different tasks exhibit pronounced alignment, which can be quantitatively assessed using the cosine similarity of singular vectors. In contrast, subspaces associated with smaller singular values demonstrate minimal overlap.
>
>   It is well understood that for vector analogies, higher alignment results in a greater norm after summation (amplify corresponding singular value in the merged model), whereas orthogonal vectors yield a smaller norm upon summation. This observation motivates an adaptive adjustment of different subspaces based on their degree of alignment.
>
> [1]. Perturbation theory for the singular value decomposition.

---

> ### Author Response · Authors · 2025-11-23
> **Response to Weakness 2**
>
> **W2: Comparison with Iso-C**
>
> * **W2.1 Similarity to Prior Work**
>
>   The Subspace Alignment discussed in this paper represents a concept that is entirely distinct from the Subspace Alignment in Iso-C. Iso-C focuses on the extent of overlap between specialized models and the merged model, whereas the **our work investigates the alignment between different specialized models**. Therefore, the analysis in Section 3.2 diverges fundamentally from that of Iso-C. Specifically, lines 180 to 190 examine the alignment between subspaces of expert models, highlighting the similarity of singular vectors across different task matrices and explaining how such similarity leads to amplification of activations in the merged model. This line of discussion is not mentioned in the Iso-C paper. In addition, arguments such as orthogonal task vectors have even already been discussed in Task Arithmetic, as cited in line 182 of this paper, and are referenced here primarily to introduce our new findings. In conclusion,
> * **W2.2 Metric Comparison (SAI vs. SAR)**
>
>   As previously discussed, our alignment focuses on the alignment among specialized models, which constitutes the principal distinction between SAR and our proposed SAI. Notably, our approach can also reveals the limitations of the SAR metric. Although our method achieves higher accuracy (*i.e.*, NAI), the corresponding SAR value is lower than that of Iso-C, indicating that the **SAR metric is not fully positively correlated with accuracy**.
>
>   The fundamental problem with the SAR metric lies in its dependence on the parameter $k_\mathrm{M}$, which can be arbitrarily controlled. According to the original definition , an increase in $k_\mathrm{M}$ leads to a higher SAR value. Based on the definition of $k_\mathrm{M}$, averaging singular values at any point leads to a higher $k_\mathrm{M}$ and consequently elevates the SAR value. For instance, consider a merging model obtained by TSV-M with a task matrix $\Delta_m$ (accuracy: 83.98%). Averaging the singular values of $\Delta_m$ produces $\Delta_m'$ (accuracy: 79.63%). Clearly, $\Delta_m'$ receives a higher SAR value with higher $k_\mathrm{M}$ than $\Delta_m$, yet yields lower accuracy (4.35%↓). In contrast, the proposed **SAI metric directly evaluates the alignment among specialized models** without introducing parameters such as $k_\mathrm{M}$ and thereby such problems as mentioned above will not occur.

---

> ### Author Response · Authors · 2025-11-23
> **Response to Weakness 2 (Continued)**
>
> * **W2.3 Difference on Top-k SVD Selection**
>
>   As stated, Iso-CTS is computed based on Task Arithmetic, whereas AlignMerge-O operates directly on each task-specific matrix; this constitutes a fundamental distinction, emphasizing that the proposed method **focuses on alignment among task-specific matrices**. Furthermore, since the top singular vector inherently captures the most crucial task-related information, this approach can be considered a common and intuitive strategy. In conclusion, although both of us find the importance of the top-k subspaces, we use the totally different operations to process them.
>
> * **W2.4 Singular Value Averaging Shared**
>
>   It should be noted that, beginning from Section 3.1 PRELIMINARY, the superscript letter K consistently denotes the total number of tasks or models (see line 146), while the subscript letter R refers to the total number of subspaces (see line 163). For instance, indicates the r-th subspace of the k-th model. In the formula for averaging provided in line 329, clearly **represents the average of the top-1 singular values across different task matrices**. In contrast, the averaging of singular values in Iso-C, according to our notation, should be expressed as $\frac{1}{R} \sum_{i=1}^R \sigma_{k}^{i}$, which averages multiple singular values within a single task matrix. Therefore, although both are averages of singular values, their target object, and the corresponding purposes are naturally completely distinct.
>
> * **W2.5 Similar Orthogonal Projection**
>
>   Orthogonal projection is a standard and straightforward operation, and the subjects of our operations are entirely distinct from Iso-CTS, exhibiting no correlation. In Eq. (12), the residual component of each specialized model's top-1 subspace, $\mathbf{M}_i^1$, is calculated with respect to the extracted shared subspace, , in order to retain unique, task-specific information after removing the components explained by the shared subspace. In contrast, Iso-C computes the residual of the complete task-specific matrix, ∆t, relative to the common subspace $\mathbf{U}^{1:r}$, which yields the residual with respect to the entire merged model rather than merely the top-1 subspace. Therefore, the objects and goals for our orthogonal projection are completely different.
>
> * **W2.6 Whitening for Bases Employed**
>
>   The whitening operation in Line 347 is explicitly cited from TSV-M in our paper. We highly respect the contribution of TSV-M. Since Iso-CTS can incorporates whitening operation from TSV-M, adopting such operation in our work is also justified.

---

> ### Comment · Reviewer_2pcv · 2025-11-26
>
> I thank the authors for answering my questions. As stated in my previous review, and as I reiterate here, the paper would benefit from a clearer contextualization of the proposed methodology with respect to prior work on subspace alignment in model merging [1]. Both the title (“Subspace-Alignment as the Missing Piece of Model Merging”), the introduction and the overall narative in the rest of the paper give the impression that the concept of Subspace Alignment in model merging is entirely novel, which is not the case.
>
> While I appreciate the authors explanations regarding the limitations of previous approaches, and I agree with some points they raise, specifically regarding the smaller eigenvector truncation; I do not agree about the type of alignment analyzed in this work is conceptually different from that in [1].  The central issue remains that the paper does not clearly articulate how the proposed method addresses or overcomes the limitations of prior work, nor how the proposed subspace alignment metrics fundamentally differ from the existing ones. Clarifying this distinction would improve the paper's contribution.
>
> More broadly, the paper remains difficult tor read. The structure, heavy notation, complex figures, and mathematical expressions that are insufficiently explained in the main text (a short description after each formula is required) make the methodology challenging to follow. Including a concise pseudocode block or a schematic figure to illustrate the overall workflow would help readers follow the methodology more easily and improve clarity. Moreover, essential definitions and explanations are deferred to the appendix. Reviewers are not expected to rely on the appendix to understand the core contribution; doing so weakens clarity and accessibility.
>
> For these reasons, I maintaing my reject score.
>
>
>  [1] Marczak, D., Magistri, S., Cygert, S., Twardowski, B., Bagdanov, A.D. & Van De Weijer, J.. (2025). No Task Left Behind: Isotropic Model Merging with Common and Task-Specific Subspaces. Proceedings of the 42nd International Conference on Machine Learning.

---

### Meta-Review · Area_Chair_ipZe · 2026-01-06

**Summary:**

The paper addresses model merging via subspace alignment and proposes two training-free methods, AlignMerge-B and AlignMerge-O, with solid empirical gains on several benchmarks. However, reviewers agree that the core contribution is not clearly distinguished from prior SVD-based and subspace-alignment methods (especially Iso-C/Iso-CTS and TSV-M), that the presentation is overly complex and hard to follow, and that several key concepts and definitions are not sufficiently self-contained in the main text. As a result, the novelty and soundness of the contribution cannot be reliably assessed in its current form. I therefore recommend rejection.

**Reviewer Concerns:**

The authors’ rebuttal clarifies some implementation details (e.g., the use of top-1 singular subspaces and layer choices) and better explains their intuition about alignment and singular value averaging. However, the main concerns remain. First, the paper still does not convincingly and systematically separate its notion of subspace alignment and its metrics from those in prior work, nor does it clearly position its methods relative to Iso-C/Iso-CTS and TSV-M beyond qualitative statements. Second, the manuscript remains difficult to read: heavy notation, insufficient explanatory text around formulas, dependence on the appendix for core definitions, and complex figures are not addressed in a way that can be judged within this review cycle. Third, comparisons to recent methods that use modest extra training or dynamic merging are not supported by quantitative efficiency analyses, so claimed advantages remain speculative. These unresolved issues underpin the reviewers’ decision to maintain their negative overall recommendations.

**Reviewer Scores:**

Reviewer 2pcv explicitly confirmed that their main concerns about originality, clarity, and the relationship to [1] were not resolved and maintained their reject score; I do not expect their score would increase after discussion. Reviewer Pmir appreciated the clarifications but reiterated concerns about the conceptual relation between alignment and conflict, and about the lack of quantitative comparisons to recent SOTA methods; they also confirmed keeping their reject recommendation, so their score would likely remain unchanged. Reviewer HBD5 saw value in the alignment perspective but asked for a much clearer distinction from prior SVD-based approaches and a more convincing explanation of the drop in Iso-C performance. Overall, there is insufficient support among reviewers to move the paper above the acceptance threshold.

---

### Decision · Program_Chairs · 2026-01-26

Reject